# Active avoidance requires inhibitory signaling in the rodent prelimbic prefrontal cortex

Maria M Diehl[1,2]*, Christian Bravo-Rivera[1,2], Jose Rodriguez-Romaguera[1,2], Pablo A Pagan-Rivera[1,2], Anthony Burgos-Robles[3], Ciorana Roman-Ortiz[1,2], Gregory J Quirk[1,2]

[1]Department of Psychiatry, University of Puerto Rico School of Medicine, San Juan, Puerto Rico; [2]Department of Anatomy & Neurobiology, University of Puerto Rico School of Medicine, San Juan, Puerto Rico; [3]Department of Brain and Cognitive Sciences, Massachusetts Institute of Technology, Cambridge, United States

**Abstract** Much is known about the neural circuits of conditioned fear and its relevance to understanding anxiety disorders, but less is known about other anxiety-related behaviors such as active avoidance. Using a tone-signaled, platform-mediated avoidance task, we observed that pharmacological inactivation of the prelimbic prefrontal cortex (PL) delayed avoidance. Surprisingly, optogenetic silencing of PL glutamatergic neurons did not delay avoidance. Consistent with this, inhibitory but not excitatory responses of rostral PL neurons were associated with avoidance training. To test the importance of these inhibitory responses, we optogenetically stimulated PL neurons to counteract the tone-elicited reduction in firing rate. Photoactivation of rostral (but not caudal) PL neurons at 4 Hz impaired avoidance. These findings suggest that inhibitory responses of rostral PL neurons signal the avoidability of a potential threat and underscore the importance of designing behavioral optogenetic studies based on neuronal firing responses.
DOI: https://doi.org/10.7554/eLife.34657.001

*For correspondence:
maria.m.diehl@gmail.com

Competing interests: The authors declare that no competing interests exist.

## Introduction

Core symptoms of post-traumatic stress disorder and other anxiety disorders include excessive fear and avoidance (*American Psychiatric Association, 2013*). The neural mechanisms of excessive fear have been well-characterized in rodents using Pavlovian fear conditioning (*Johansen et al., 2011*; *Duvarci and Pare, 2014*; *Herry and Johansen, 2014*; *Giustino and Maren, 2015*; *Do Monte et al., 2016*), yet the mechanisms of active avoidance are just beginning to emerge. Previous work in rats has shown that the prefrontal cortex, amygdala, and striatum are all necessary for the expression of active avoidance (*Martinez et al., 2013*; *Moscarello and LeDoux, 2013*; *Beck et al., 2014*; *Jiao et al., 2015*; *LeDoux et al., 2017*). Using a tone-signaled, platform-mediated avoidance task, we observed that pharmacological inactivation of the prelimbic prefrontal cortex (PL) impaired the expression of avoidance without affecting freezing (*Bravo-Rivera et al., 2014*). Furthermore, avoidance that persisted following extinction was correlated with excessive PL activity, as indicated by the immediate early gene cFos (*Bravo-Rivera et al., 2015*), suggesting that PL activity may drive the expression of active avoidance.

Important questions remain, however, regarding the role of PL in avoidance. First, how do PL neurons signal avoidance? Fear conditioning mainly induces excitatory responses to conditioned tones in PL that correlate with freezing (*Baeg et al., 2001*; *Burgos-Robles et al., 2009*; *Sotres-Bayon et al., 2012*; *Isogawa et al., 2013*; *Pendyam et al., 2013*; *Chang et al., 2010*), but the firing properties of PL neurons in active avoidance have not been studied. In platform-mediated

avoidance, PL signaling of avoidance may differ from PL signaling of freezing or foraging for food (*Burgos-Robles et al., 2013*), both of which can interfere with platform avoidance. Second, does avoidance involve all of PL or only specific subregions?

We addressed these questions by recording PL neurons during tone-signaled, platform-mediated avoidance. We then optogenetically silenced or activated PL neurons based on the observed firing patterns. We found that inhibitory (rather than excitatory) tone responses of rostral PL neurons were associated with avoidance. Opposing these inhibitory responses with photoactivation delayed or prevented active avoidance, suggesting that prefrontal inhibition signals the 'avoidability' of danger.

## Results

### Pharmacological inactivation of PL delays avoidance

We first replicated our prior findings that pharmacological inactivation of PL with the GABA-A agonist muscimol (MUS) impaired avoidance in this task (*Bravo-Rivera et al., 2014*), with two modifications: (1) we used fluorescently labeled MUS to assess spread to adjacent regions, and (2) we analyzed the time course of avoidance behavior across the 30 s tone. Because the 2 s shock co-terminates with the tone, the rat has 28 s to stop pressing the lever for food and step onto the platform to escape the shock. Furthermore, in this task, avoidance comes at a cost, as it competes with access to food. Thus, the involvement of PL could vary with changes in the cost and/or urgency of avoidance as the tone progresses (*Zeeb et al., 2015*; *Hosking et al., 2016*).

Histological analysis showed that MUS was confined to PL in its mid rostral-caudal extent (*Figure 1A*). Rats with substantial spread to adjacent infralimbic cortex were excluded (n = 3). In some cases, MUS reached the ventral half of cingulate cortex (Cg1), and these cases were included due to similar functions of Cg1 and PL in conditioned fear (*Courtin et al., 2014*) and avoidance (*Orona and Gabriel, 1983*; *Freeman et al., 1996*). Following surgical implantation of cannulas, rats were trained in platform-mediated avoidance over 10 days as previously described (*Figure 1B*, *Bravo-Rivera et al., 2014*; *Rodriguez-Romaguera et al., 2016*). On Test 1 (Day 11), we infused MUS into PL at the same concentration as our prior studies using fluorescent MUS (*Do-Monte et al., 2015b*; *Rodriguez-Romaguera et al., 2016*) and waited 45 min before commencing a 2-tone test of avoidance expression (without shock). *Figure 1C* shows that MUS inactivation significantly reduced the time spent on the platform during the tone, as compared to saline (SAL) infused controls (SAL 92% vs. MUS 57%, $t_{(28)} = -4.019$, p<0.001, Bonferroni corrected). An analysis of avoidance across the tone in 3 s bins (*Figure 1D*) indicated that MUS-infused rats were significantly delayed in their initiation of avoidance (repeated measures ANOVA, $F_{(1,9)} = 4.076$, p<0.001; post hoc, 0–15 s **p<0.01, 15–21 s *p<0.05), and 2/13 rats never avoided (Mann Whitney U Test, p<0.001, *Figure 1E*). MUS also increased tone-induced freezing (*Figure 1E* top inset; SAL = 36% vs. MUS = 55% freezing, $t_{(28)} = 2.460$, p=0.020) but had no effect on suppression of bar pressing (*Figure 1E* bottom inset; SAL = 0.922 vs. MUS = 0.984 suppression ratio, $t_{(28)} = 0.194$, p=0.848). Inactivation of PL had no effect on locomotion, as indicated by distance traveled during a 5 min open field test (SAL n = 10, 13.23 m vs. MUS n = 10, 12.53 m, $t_{(18)} = 0.513$, p=0.614, *Figure 1—figure supplement 1*). Nor was there an effect on anxiety, as assessed with time spent in the center of the open field (SAL = 15.69 s vs. MUS = 18.76 s, $t_{(18)} = 0.933$, p=0.363, *Figure 1—figure supplement 1*). Thus, pharmacological inactivation of PL delayed the expression of active avoidance.

### Photosilencing of PL glutamatergic neurons does not delay avoidance

Because pharmacological inactivation of PL delayed avoidance, we reasoned that tone-induced activity in PL would be essential for avoidance early in the tone. To assess this, we used an optogenetic approach, expressing the microbial opsin archaerhodopsin (ArchT) in PL, which causes a hydrogen proton efflux to hyperpolarize neurons when exposed to 532 nm (green) light (*Chow et al., 2010*; *Han et al., 2011*). We delivered ArchT by infusing an adeno-associated virus (AAV) encoding both ArchT and enhanced yellow fluorescent protein (eYFP) under the control of the CAMKIIα promoter to target glutamatergic neurons (*Jones et al., 1994*, AAV5:CaMKIIα::eArchT3.0-eYFP; *Liu and Jones, 1996*, *Van den Oever et al., 2013*, *Warthen et al., 2016*). We first confirmed in anesthetized rats that ArchT silences PL neurons by recording extracellular activity from ArchT-infused rats exposed to green light (*Figure 2A*). Laser illumination significantly decreased the firing

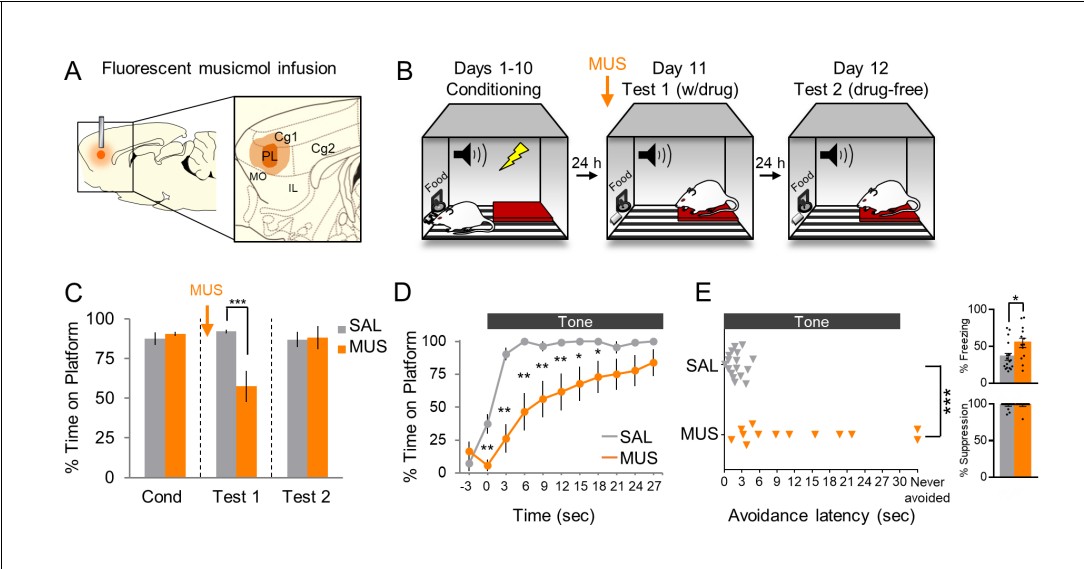

**Figure 1.** Pharmacological inactivation of prelimbic cortex delays avoidance. (**A**). Schematic of MUS infusion showing the minimum (dark orange) and maximum (light orange) extent of infusion into PL. (**B**). Rats were trained across 10 days to avoid a tone-signaled foot-shock by stepping onto a platform. On Day 11, rats received two tone presentations (without shock) 45 min after MUS infusion. On Day 12, rats received a second 2-tone test drug free. (**C**). Percent time on platform during Tone 1 on Days 10, 11, and 12 for MUS and saline controls (SAL, n = 17; grey) and MUS rats (n = 13, orange). (**D**). Time spent on platform in 3 s bins for Test 1 (Tone 1) revealed that MUS rats were significantly delayed in their avoidance compared to SAL controls (repeated measures ANOVA, post hoc Tukey). (**E**). Latency of avoidance for each rat (Mann Whitney U test, Tone 1, Test 1). *Inset*: Effect of MUS inactivation (Tone 1, test 1) on freezing (top) and percent suppression of bar pressing (bottom) during the tone (unpaired t-test). Data are shown as mean ± SEM; *p<0.05, **p<0.01, ***p<0.001.

DOI: https://doi.org/10.7554/eLife.34657.002

The following source data and figure supplement are available for figure 1:

**Source data 1.** Open field measures following MUS infusion in PL.
DOI: https://doi.org/10.7554/eLife.34657.004

**Figure supplement 1.** Assessment of locomotion and anxiety following pharmacological inactivation of PL.
DOI: https://doi.org/10.7554/eLife.34657.003

rate of 38/70 neurons and increased the firing rate of 9/70 neurons (Wilcoxon signed-ranks test comparing pre-laser vs laser activity of each unit using 1 s time bins, all p's <0.05).

Next, we infused ArchT bilaterally into PL, distinguishing rostral PL (rPL; defined as dorsal to medial orbitofrontal cortex and anterior to the infralimbic cortex) from caudal PL (cPL; defined as dorsal to the infralimbic cortex; *Figure 2B*) based on distinct connectivity of these subregions (*Floyd et al., 2000*; *Floyd et al., 2001*). 4–6 weeks after viral infusion, 10 days of avoidance training commenced. Rats were then given a 2-tone test of avoidance expression, with laser illumination concurrent with the first tone only. Surprisingly, avoidance was not impaired by photosilencing of either rPL (*Figure 2C*; $t_{(30)}$ = 0.792, p=0.435) or cPL (*Figure 2D*; $t_{(14)}$ = 0.471, p=0.646). Photosilencing also had no effect on the time course of avoidance in rPL (*Figure 2E* left) or cPL (*Figure 2F* left). However, rPL-ArchT rats avoided significantly *earlier* than eYFP controls, as measured by avoidance latency (*Figure 2E* right, Mann Whitney U test, p=0.021). With respect to freezing, there was no significant effect of photosilencing in rPL (*Figure 2C* top inset, $t_{(30)}$ = 1.939, p=0.062) or cPL (*Figure 2D* top inset, $t_{(14)}$ = 0.590, p=0.565). Suppression of bar pressing was also unaffected by photosilencing in either location (*Figure 2C* bottom inset, rPL: $t_{(30)}$ = 0.415, p=0.681; *Figure 2D* bottom inset, cPL: $t_{(14)}$ = 0.984, p=0.342). The lack of impairment of avoidance may suggest that we failed to sufficiently inhibit PL activity via ArchT photosilencing. However, photosilencing rPL neurons during early avoidance training (on day 2) significantly reduced tone-induced freezing (eYFP-control: 31% (n = 9) vs. eYFP-ArchT: 7% (n = 8), $t_{(15)}$ = 0.288, p=0.012, *Figure 2—figure supplement 1*).

Thus, contrary to our initial hypothesis, excitatory activity of PL projection neurons does not appear to be necessary for avoidance behavior. Instead, silencing rPL tended to facilitate avoidance

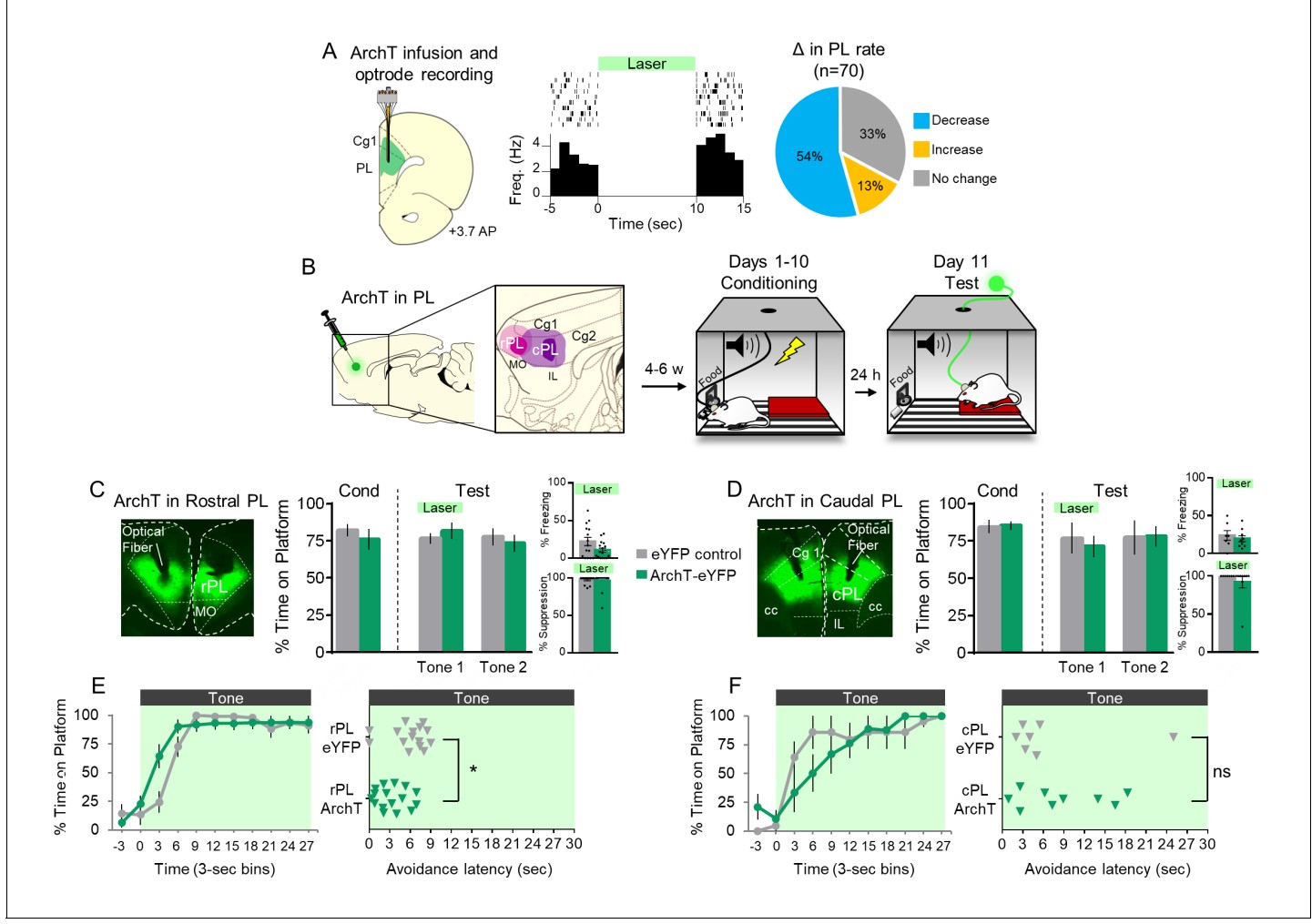

**Figure 2.** Optogenetic silencing of prelimbic neurons does not delay avoidance. (**A**). *Left*: Schematic of ArchT expression and optrode placement in anesthetized rats (n = 2). *Middle*: Rasters and peristimulus time histogram of a single PL neuron showing a decrease in firing rate during laser illumination (8–10 mW, 532 nm, 10 s ON, 10 s OFF, 10 trials). *Right*: Proportion of PL neurons that exhibited a decrease (blue, n = 38), increase (gold, n = 9), or no change (grey, n = 23) in firing rate. (**B**). Schematic of virus infusion, location of min/max expression of AAV in rPL (pink) and cPL (purple), followed by avoidance training and test. At Test, 532 nm light was delivered to rPL or cPL during the entire 30 s tone presentation (Tone 1). (**C**). *Left*: Micrograph of ArchT expression and optical fiber placement in rPL. *Right*: Percent time on platform at Cond (Day 10, Tone 1) and Test (Day 11, Tone 1 with laser ON and Tone 2 with laser OFF) for rPL-eYFP control (n = 15, grey) and rPL-ArchT rats (n = 17, green). *Inset*: There was no effect of rPL photosilencing (Tone 1 at Test) on freezing (top) and percent suppression of bar pressing (bottom) during the tone (unpaired t-test). (**D**). *Left*: Micrograph of ArchT expression and optical fiber placement in cPL. *Right*: Percent time on platform during Cond and Test for cPL-eYFP control (n = 7, grey) and cPL-ArchT rats (n = 9, green). *Inset*: There was no effect of cPL photosilencing (Tone 1 at Test) on freezing (top) and percent suppression of bar pressing (bottom) during the tone (unpaired t-test). (**E**). *Left*: Time spent on platform in 3 s bins (Tone 1 at Test) revealed no effect of silencing rPL-ArchT neurons compared to eYFP controls (repeated measures ANOVA). *Right*: Latency of avoidance for each rat (Tone 1 at Test). rPL-ArchT rats showed a decrease in avoidance latency (Mann Whitney U test, p=0.021). (**F**). Timeline of avoidance (left) and latency (right) for cPL-eYFP control rats and cPL-ArchT rats. All data are shown as mean ± SEM; *p<0.05.

DOI: https://doi.org/10.7554/eLife.34657.005

The following source data and figure supplement are available for figure 2:

**Source data 1.** Freezing levels following ArchT silencing of rPL neurons.
DOI: https://doi.org/10.7554/eLife.34657.007
**Figure supplement 1.** Assessment of fear following ArchT silencing of rPL neurons.
DOI: https://doi.org/10.7554/eLife.34657.006

(as indicated by the decrease in avoidance latency), raising the possibility that avoidance signaling may involve rPL inhibition rather than excitation.

## Inhibitory tone responses of PL neurons are specific to avoidance

An assumption of our photosilencing approach was that increased activity in PL neurons is correlated with avoidance; however, this hypothesis had never been tested. We therefore performed extracellular single unit recordings in PL of well-trained rats during avoidance expression. Units were recorded from the full rostral-caudal extent of PL (*Figure 3A*). We first characterized PL responses to tone onset. Both excitatory responses (Z > 2.58, first 500 ms) and inhibitory responses (Z < −1.96, in the first or second 500 ms) were observed (*Figure 3B* right). This tone response latency (<1 s) was selected to ensure that the activity of PL neurons reflected the tone rather than platform entry, which occurred later than 1 s in 91% of the trials (median = 3.55 s). *Figure 3C* shows the proportions of neurons that were significantly responsive (at each 500 ms bin) throughout the tone. The black dots above the graph indicate the time of platform entry relative to tone onset. Out of 205 neurons, 30 were excited (14%) and 22 were inhibited (11%) at tone onset, relative to 10 s of pre-tone activity (*Figure 3D*). Normalized activity throughout the tone for all neurons is shown in *Figure 3—figure supplement 1A–B*.

To determine if these tone responses were correlated with avoidance rather than simply auditory processing, we compared PL responses in this group of rats with those of a naïve control group trained to press for food and presented with tones in the same chamber with the platform. Naïve rats were free to mount the platform and explore the chamber but were never shocked. In addition, to determine whether activity at tone onset might represent the conditioned aversiveness of the tone, we compared responses in avoidance rats with responses in rats subjected to auditory fear conditioning in the same chamber (re-analysis of data from *Burgos-Robles et al., 2009*). Surprisingly, there were no significant differences in the percentage of excitatory tone responses in the avoidance group compared to the naïve or fear conditioned groups (*Figure 3D* top right; avoidance-trained: 30/205 (14%), naïve: 20/166 (12%), fear: 25/191 (13%), Chi Square = 0.547, p=0.761). Inhibitory responses, however, occurred more frequently in avoidance-trained rats compared to the other two groups (avoidance-trained: 22/205 (11%), naïve: 3/166 (2%), fear: 3/191 (2%), Chi Square = 22.545, p<0.001). Group differences between tone responses are shown for the first 5 s of the tone in *Figure 3D* (bottom). Note the marked differences between avoidance and naïve groups for inhibitory, but not excitatory, responses at tone onset.

## Platform entry responses are not specific to avoidance

We next examined PL activity at platform entry, defined as the moment at which the rat's head entered the platform zone (*Figure 3E–G*), compared to the same baseline used for tone onset. Both excitatory (Z > 2.58 in the first 500 ms) and inhibitory (Z < −1.96 in the first or second 500 ms) responses to platform entry were observed (*Figure 3E* right). *Figure 3F* shows the proportion of neurons that were responsive at each 500 ms time bin around platform entry (black dots above the graph show tone onsets). PL neurons showed excitation (n = 26/175; 15%) and inhibition (n = 16/175; 9%) at platform entry (*Figure 3G* left), but neither differed significantly from the naïve group (*Figure 3G* right: n = 23/160 excited, p=0.331; n = 10/160 inhibited, p=0.197 Fisher Exact). Platform responses across the first 5 s after platform entry are shown in *Figure 3G* (bottom). Cells showing excitatory responses to the tone were largely distinct from cells showing excitatory responses to platform entry, but there was some overlap between responses showing inhibition (*Figure 3H*). Together, these results suggest that responses to platform entry represent sensory perception and/or motor responses rather than avoidance of threat (*Amir et al., 2015*).

We next asked if the latency of PL inhibition to the tone correlated with the latency of platform entry. Inhibition latency was defined as the start of the first inter-spike interval (ISI) that was significantly longer than the average pre-tone ISI (Z > 1.65; p<0.05). 133/205 neurons showed at least one ISI that satisfied this criterion. The latency of inhibition showed no correlation with the latency of platform entry (r = 0.022, Pearson correlation, *Figure 3—figure supplement 1E*). For each cell, we averaged its inhibitory latency across all the trials in which successful avoidance was observed (out of nine trials in each session, n = 284 trials), as well as the avoidance latency on those trials. The inhibitory response in most cells preceded platform entry (88/133 cells) but was not correlated with the

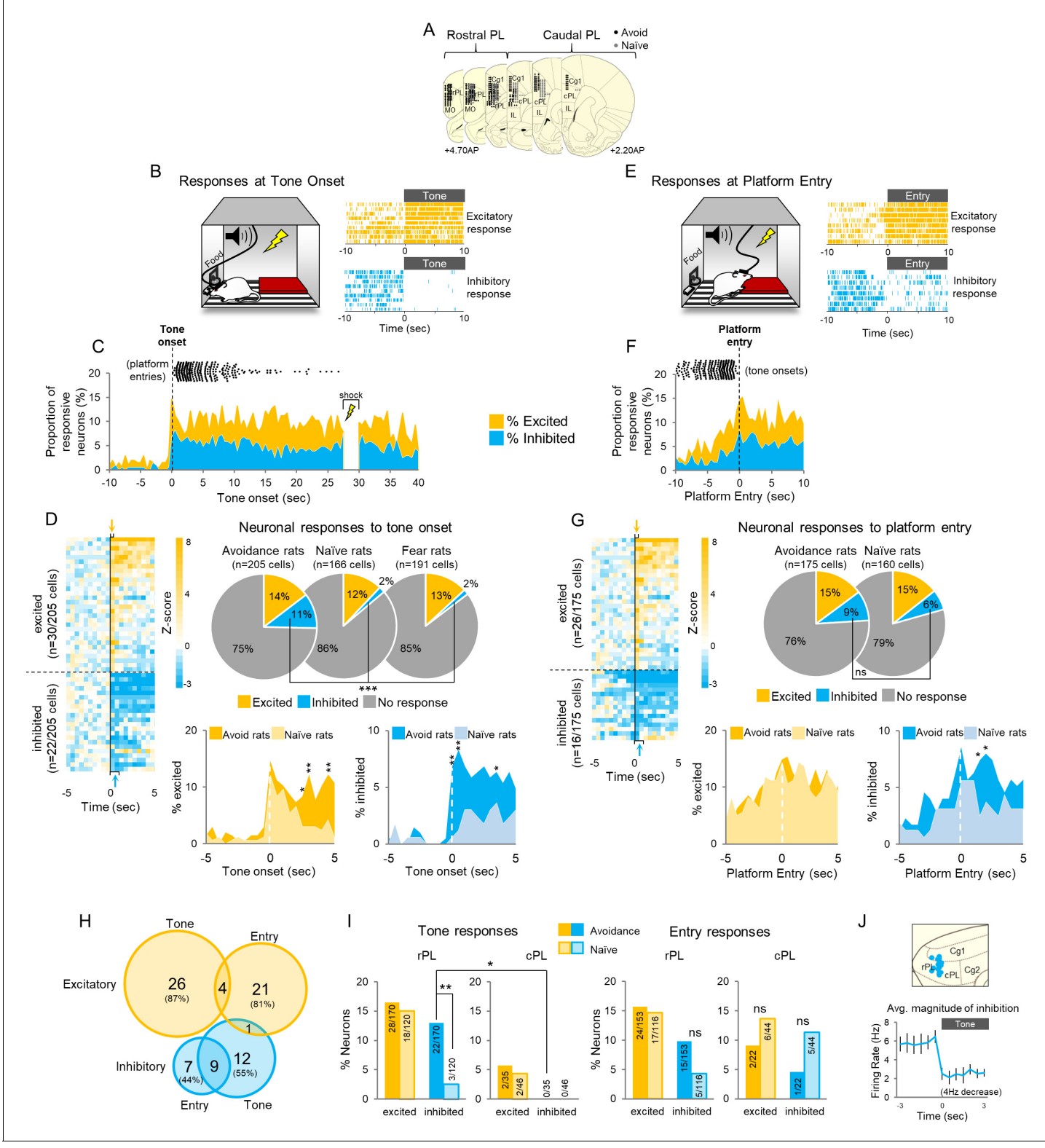

**Figure 3.** Active avoidance is correlated with inhibition in rostral PL neurons. (**A**). Location of recordings across PL (n = 6 avoidance-trained and n = 8 naïve rats). (**B**). *Left*: Schematic of rat behavior at tone onset during unit recordings. *Right*: single unit examples of excitatory (gold rasters) and inhibitory (blue rasters) tone responses. Each row represents a single trial. (**C**). Proportion of excitatory (gold) or inhibitory (blue) neurons at each 500 ms bin across the tone. Time of platform entry (black dots), for all successful trials (n = 284) in avoidance rats is indicated relative to tone onset. (**D**). *Left*: Heat map of normalized (z-score) responses to tone onset (Time = 0 s) of neurons in avoidance rats. Each row represents one neuron, bin = 0.5 s. Arrows

*Figure 3 continued on next page*

*Figure 3 continued*

indicate bins used to determine excitatory (gold, first 500 ms bin), or inhibitory (blue, first or second 500 ms bin) tone responses. *Right:* Pie charts showing proportions of excited, inhibited, or non-responsive neurons at tone onset in avoidance (n = 30, 22, 153, respectively), naïve (n = 20, 3, 143, respectively), and fear conditioned (n = 25, 3, 163, respectively) rats. Proportions of inhibitory responses were significantly greater in avoidance rats compared to naïve and fear conditioned rats (Chi Square test). *Bottom:* Percentage of cells that were excited in avoidance (gold) or naïve (light gold) rats (left) or inhibited in avoidance (blue) or naïve (light blue) rats (right) around tone onset (Fisher exact tests). (E). *Left:* Schematic of rat entering platform after tone onset during unit recordings. *Right:* single unit examples of excitatory (gold rasters) and inhibitory (blue rasters) platform entry responses. (F). Proportion of excitatory (gold) or inhibitory (blue) neurons at platform entry. Time of tone onset (black dots), for all successful trials (n = 284) in avoidance rats is indicated relative to platform entry. (G). *Left:* Heat map of normalized responses to platform entry (Time = 0 s) of neurons in avoidance rats. *Right:* Pie charts showing proportions that were excited, inhibited, or non-responsive neurons at platform entry in avoidance (n = 26, 16, 133, respectively) and naïve rats (n = 23, 10, 127, respectively). *Bottom:* Percentage of cells that were excited in avoidance (gold) or naïve (light gold) rats (left) or inhibited in avoidance (blue) or naïve (light blue) rats (right) after platform entry (Fisher exact tests). (H). Venn diagram illustrating the number (and percentage) of excitatory and inhibitory responsive cells responding to tone onset, platform entry, or both. (I). *Left:* Proportion of neurons responding to tone onset in rostral PL (left) and caudal PL (right) in avoidance (dark bars) and naïve (light bars) groups. There were significantly more inhibitory tones responses in rPL vs cPL (Fisher Exact test). *Right:* Proportion of neurons responding to platform entry in rostral PL (left) and caudal (right) PL in avoidance and naïve rats. (J). *Top:* Sagittal view of location of inhibitory tone responsive neurons (blue). *Bottom:* Average inhibitory response of neurons decreased from a baseline firing rate of 5.8 Hz to 1.98 Hz at tone onset. Data are shown as mean ± SEM; *p<0.05, **p<0.01, ***p<0.001.

DOI: https://doi.org/10.7554/eLife.34657.008

The following source data and figure supplement are available for figure 3:

**Source data 1.** PL unit recording data.
DOI: https://doi.org/10.7554/eLife.34657.010
**Figure supplement 1.** Characterization of PL single unit responses during avoidance.
DOI: https://doi.org/10.7554/eLife.34657.009

latency of platform entry (r = 0.078, *Figure 3—figure supplement 1F*). In fact, similar inhibition was observed in trials where the rat chose not to avoid (n = 107 trials, dashed orange line in *Figure 3—figure supplement 1F*). The latency of headturn, which was the first movement the rat made before proceeding to the platform, also did not correlate with the latency of inhibitory responses (*Figure 3—figure supplement 1E*). Rather than signaling avoidance behavior, therefore, inhibitory responses in PL appear to signal that shock can be avoided (an avoidance option), regardless of whether the rat chose to avoid on that trial.

## Opposing inhibition within rostral PL delays or prevents avoidance

Further analysis revealed that all neurons showing inhibition to the tone were located in rPL (blue, n = 22), with none in cPL (*Figure 3I–J*). Most inhibitory responses (n = 18/22) were brief, ending by ~10 s after tone onset, whereas a smaller proportion were sustained throughout the tone (n = 4/22, *Figure 3—figure supplement 1C*). Neurons showing inhibition reduced their firing rate from 6 to 2 Hz on average (*Figure 3J*) and were putative projection neurons based on their spike width and baseline firing rate (>225 µs, <15 Hz, from our previous study of PL neurons; *Sotres-Bayon et al., 2012*, see *Figure 3—figure supplement 1D*).

If inhibition within rPL signals the avoidability of a tone-signaled shock, we reasoned that opposing this inhibition should remove this option and impair avoidance. To oppose inhibition, we used channelrhodopsin (ChR2) targeting CAMKIIα-positive neurons to activate rPL neurons throughout the tone at 4 Hz, to counter the tone-induced decrease from 6 to 2 Hz. To validate our method, we first measured extracellular unit activity in anesthetized rats from ChR2-expressing rPL neurons exposed to blue light (473 nm, *Figure 4A*). *Figure 4B* shows a representative rPL neuron increasing its firing rate with photoactivation. We found that 4 Hz photoactivation increased the firing rate in 38% of the neurons and decreased the firing rate in 24% of the neurons (*Figure 4C* left; n = 112, 4 Hz, 30 s duration, 5 ms pulse width, 8–10 mW illumination, p<0.05). Photoactivation induced less than 4 Hz activity (3.33 Hz) suggesting that neurons failed to respond to some light pulses (*Figure 4C* right), as has previously been observed for ChR2 (*Warden et al., 2012*). Photoactivation at 2 Hz had an even weaker effect, increasing the firing rate from 0.4 to 1.19 Hz on average (*Figure 4D–E*).

We next infused ChR2 bilaterally into either the rPL or cPL and began avoidance conditioning 3–4 weeks after AAV infusion (*Figure 5A*). Following 10 days of avoidance training, rats were exposed to

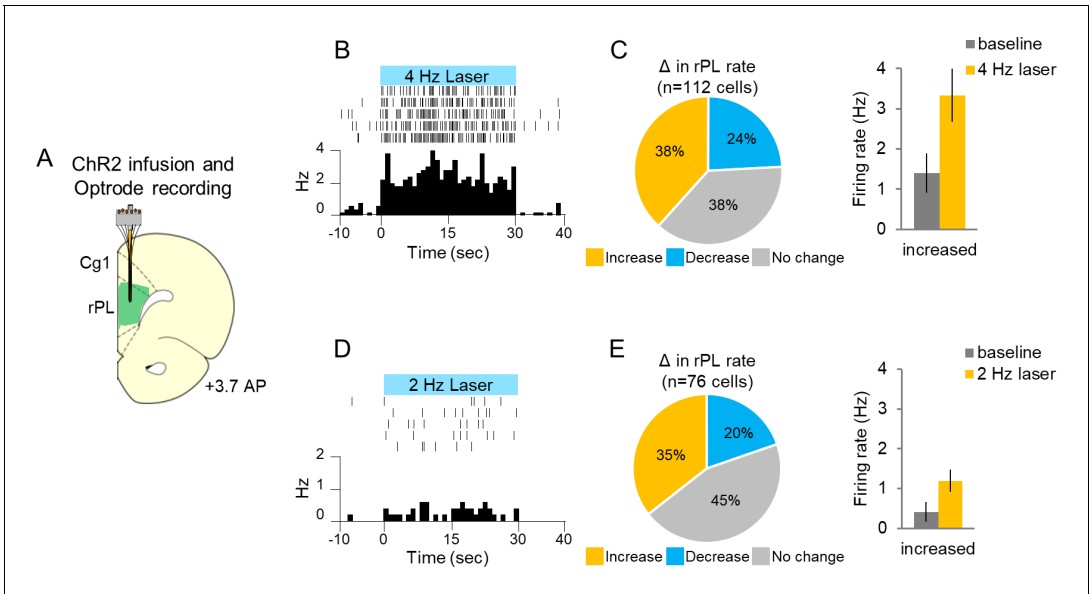

**Figure 4.** Single-unit recording with photoactivation in rostral PL neurons of anesthetized rats. (**A**). Schematic of ChR2 expression and optrode placement (n = 4 rats). (**B**). Rasters and peristimulus time histograms of a representative single neuron showing increased firing rate during 4 Hz laser illumination (8–10 mW, 473 nm, 30 s ON, 30 s OFF, five trials). (**C**). *Left*: Proportion of neurons showing an increase (gold, n = 43), decrease (blue, n = 27), or no change (grey, n = 42) in firing rate with laser ON. *Right*: Average firing rate at baseline (dark grey) and 4 Hz photoactivation for neurons showing increased (gold) changes in firing rate. (**D**). Rasters and peristimulus time histograms of a representative single neuron showing increased firing rate during 2 Hz laser illumination (8–10 mW, 473 nm, 30 s ON, 30 s OFF, five trials). (**E**). *Left*: Proportion of neurons showing an increase (n = 27), decrease (n = 15), or no change (n = 34) in firing rate with laser ON. *Right*: Average firing rate at baseline, and 2 Hz photoactivation for neurons showing increased changes in firing rate. Data are shown as mean ± SEM.
DOI: https://doi.org/10.7554/eLife.34657.011

The following source data is available for figure 4:

**Source data 1.** ChR2 anesthetized unit recording data.
DOI: https://doi.org/10.7554/eLife.34657.012

two tones presented in the absence of shock. PL neurons were illuminated throughout the first tone (4 Hz, 30 s). Photoactivation of rPL neurons at 4 Hz markedly reduced avoidance expression as reflected in the time spent on the platform (*Figure 5B*; eYFP-control, n = 9, 87% vs. ChR2-eYFP, n = 14, 27%, $t_{(21)} = -4.779$, p<0.001, Bonferroni corrected; see *Video 1*). In contrast to rPL, photoactivation of cPL had no significant effect on avoidance (Figure 5C, $t_{(14)} = 1.531$, p=0.148) or its time course (*Figure 5E*).

Examination of the time course of avoidance showed that photoactivation of rPL significantly reduced avoidance throughout the tone (*Figure 5D* left; repeated measures ANOVA, main effect (Group), $F_{(1)} = 18.642$, p<0.001, interaction effect (Group x Time) $F_{(9)} = 1.156$, p=0.326, post hoc, 3–30 s, all p's < 0.01). Photoactivation delayed avoidance in 7/14 rats and blocked avoidance entirely in 7/14 rats (*Figure 5D* right; Mann Whitney U test, p<0.001). Photoactivation had no significant effect on freezing or suppression of bar pressing in either rPL (*Figure 5D* insets, freezing: $t_{(21)} = 1.121$, p=0.275; suppression: $t_{(21)} = 1.343$, p=0.194) or cPL (*Figure 5E* insets, freezing: $t_{(14)} = 0.0702$, p=0.494; suppression: $t_{(14)} = 0.483$, p=0.636). Photoactivation at 2 Hz had no effect on avoidance expression (*Figure 5F*). Furthermore, shifting the 4 Hz photoactivation to the inter-tone interval did not impair avoidance (*Figure 5G*). Thus, the photoactivation-induced impairment of avoidance showed specificity with respect to location, time, and frequency. Finally, reducing the duration of 4 Hz photoactivation to the first 15 s of the tone delayed, but did not prevent, avoidance as indicated by time on platform (*Figure 5H* left; Mann Whitney U test, p's <0.05 at 9–15 s) and avoidance latency (*Figure 5H* right, $t_{(17)} = 3.363$, p=0.004). 4 Hz photoactivation of rPL had no effect on locomotion, as indicated by distance traveled in an open field (eYFP-control n = 11, 2.71 m vs. ChR2-eYFP, n = 15, 2.25 m, $t_{(24)} = 0.941$, p=0.356, *Figure 5—figure supplement 1*). Nor did it have any effect on anxiety levels, as assessed by time spent in the center of the open field (eYFP-

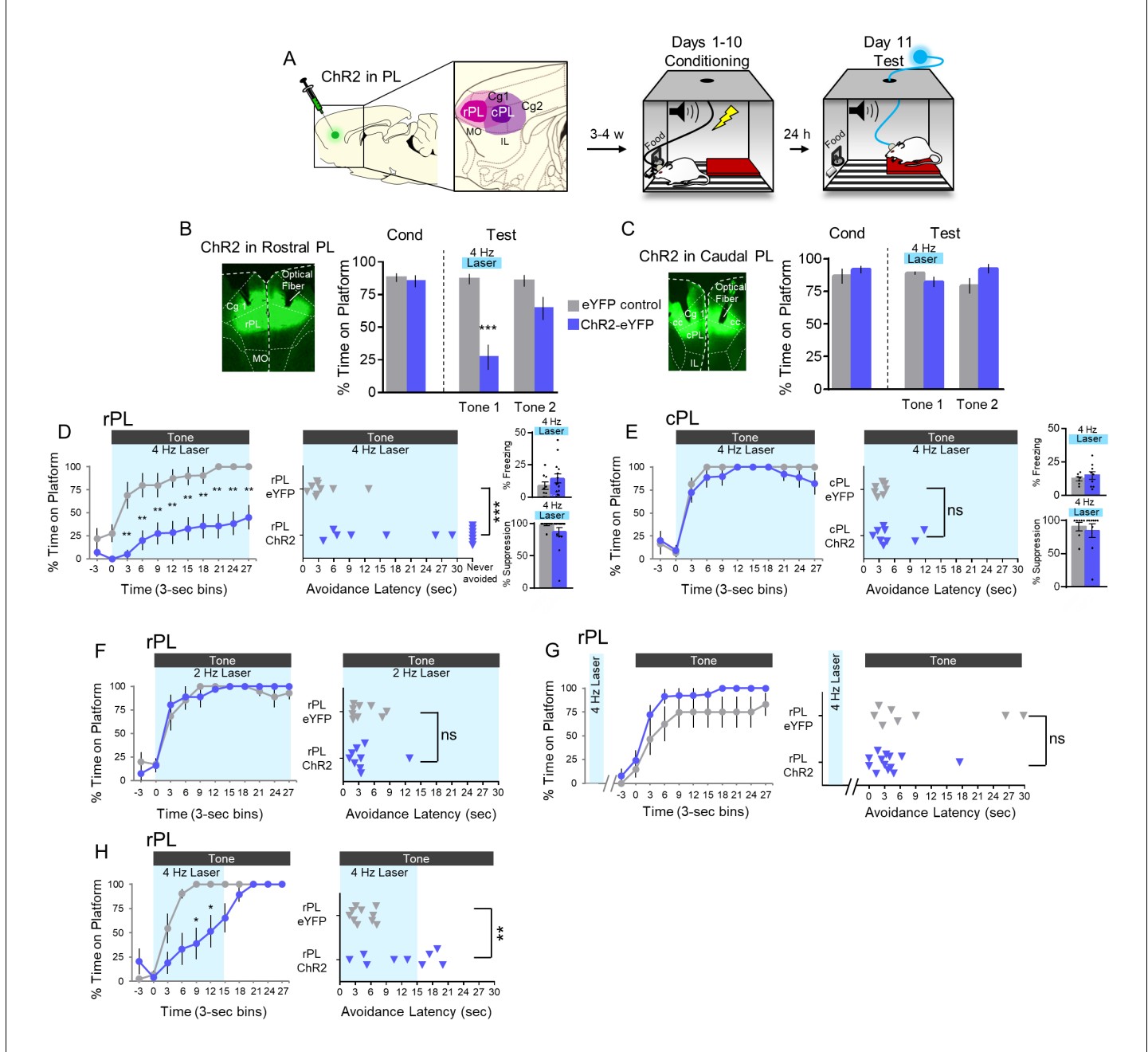

**Figure 5.** 4 Hz photoactivation of neurons in rostral PL delays or prevents avoidance. (**A**). Schematic of viral infusion and location of min/max spread of AAV expression in rPL (pink) and cPL (purple), followed by avoidance training. At Test, 473 nm light was delivered to rPL or cPL during the 30 s tone presentation (Tone 1). (**B**). *Left*: Micrograph of ChR2 expression and optical fiber placement in rPL. *Right*: Percent time on platform at Cond (Day 10, Tone 1) and Test (Day 11, Tone 1 with laser ON and Tone 2 with laser OFF) for rPL-eYFP control rats (grey, n = 9) and rPL-ChR2 rats (blue, n = 14). (**C**). *Left*: Micrograph of ChR2 expression and optical fiber placement in cPL. *Right*: Percent time on platform during Cond and Test for cPL-eYFP control rats (grey, n = 7) and cPL-ChR2 rats (blue, n = 9). (**D**). *Left*: Time spent on platform in 3 s bins (Tone one at Test) revealed that rPL-ChR2 rats were significantly delayed in their avoidance compared to eYFP controls (repeated measures ANOVA, post hoc tukey). *Right*: Latency of avoidance for each rat (Mann Whitney U test, Tone 1 at Test). 7/14 rats never avoided. *Inset*: There was no effect of rPL photoactivation (Tone 1 at Test) on freezing (top) and percent suppression of bar pressing (bottom) during the tone (unpaired t-test). (**E**). Timeline of avoidance (left) and latency (right) for ChR2-cPL rats and eYFP controls revealed no effect of 4 Hz photoactivation of cPL. *Inset*: There was no effect of cPL photoactivation (Tone 1 at Test) on freezing (top) and percent suppression of bar pressing (bottom) during the tone (unpaired t-test). (**F**). Timeline of avoidance (left) and latency (right) for rPL-ChR rats (blue, n = 9) and rPL-eYFP controls (grey, n = 9) revealed no effect of 2 Hz photoactivation. (**G**). Timeline of avoidance (left) and latency (right) for rPL-ChR2 rats (blue, n = 13) and rPL-eYFP controls (grey, n = 8) revealed no effect of 4 Hz photoactivation (30 s) during the ITI period. (**H**). Timeline of

*Figure 5 continued on next page*

*Figure 5 continued*

avoidance (left) and latency (right) for and rPL-ChR2 rats (blue, n = 9) and rPL-eYFP controls (grey, n = 10) revealed a delay in avoidance with 4 Hz photoactivation during the first 15 s of the tone (Mann Whitney U test for time course and avoidance latency). All data are shown as mean ± SEM; *p<0.05; **p<0.01; ***p<0.001.

DOI: https://doi.org/10.7554/eLife.34657.013

The following source data and figure supplement are available for figure 5:

**Source data 1.** Open field measures during blue laser illumination in rPL with ChR2.

DOI: https://doi.org/10.7554/eLife.34657.015

**Figure supplement 1.** Assessment of locomotion and anxiety following 4 Hz photoactivation of rPL neurons.

DOI: https://doi.org/10.7554/eLife.34657.014

control = 2.6727 s vs. eYFP-ChR2 = 2.6733 s, $t_{(24)}$ = 4.82e-4, p=0.999, *Figure 5—figure supplement 1*). Thus, preventing inhibition in rPL glutamatergic neurons severely impaired avoidance expression.

## Discussion

In this study, we investigated the mechanisms of prefrontal control over active avoidance. Whereas pharmacological inactivation of PL delayed avoidance, optogenetic silencing of rostral PL accelerated avoidance. Single-unit recordings revealed that avoidance training was associated with inhibitory, rather than excitatory, tone responses in rostral PL neurons. Consistent with this, opposing tone-induced inhibition by optogenetically activating rPL neurons delayed or prevented avoidance. These findings add to a growing body of evidence that inhibition within PL is key for conditioned behavior (*Ehrlich et al., 2009*; *Ciocchi et al., 2010*; *Sotres-Bayon et al., 2012*; *Sparta et al., 2014*) and highlight the importance of using in vivo recordings to guide optogenetic behavioral manipulations.

Previous work has shown that lesions or inactivation of PL reduces freezing in Pavlovian fear conditioning tasks (*Baeg et al., 2001*; *Vidal-Gonzalez et al., 2006*; *Sierra-Mercado et al., 2011*). In our platform-mediated avoidance task, however, there was no reduction in freezing following pharmacological or optogenetic inhibition of PL (present study; *Bravo-Rivera et al., 2014*). Thus, PL activity is no longer necessary for freezing following avoidance training. It is therefore unlikely that inhibitory responses in PL promote avoidance by decreasing freezing. In fact, freezing levels increased following MUS inactivation, consistent with loss of avoidance as a possible response to the tone. It is well-established that early stages of avoidance training depend on Pavlovian conditioning (acquisition of tone-shock association), whereas later stages of training shift to instrumental learning (platform entry; *Mowrer and Lamoreaux, 1946*; *Kamin et al., 1963*; *LeDoux et al., 2017*). In agreement with this shift, we did, in fact, observe a decrease in freezing following optogenetic silencing of PL early in avoidance training.

We observed both excitatory and inhibitory signaling in PL during avoidance. Excitatory responses to platform entry are consistent with prior cFos studies showing that active avoidance is correlated with increased PL activity (*Martinez et al., 2013*; *Bravo-Rivera et al., 2015*). Inhibitory responses to the tone were observed following avoidance training, but not fear conditioning, suggesting that inhibition is specific to avoidance. However, inhibitory tone responses were not correlated with platform entry and persisted in trials in which the rat did not avoid. Instead of signaling avoidance behavior, we suggest that rPL inhibition is a training-

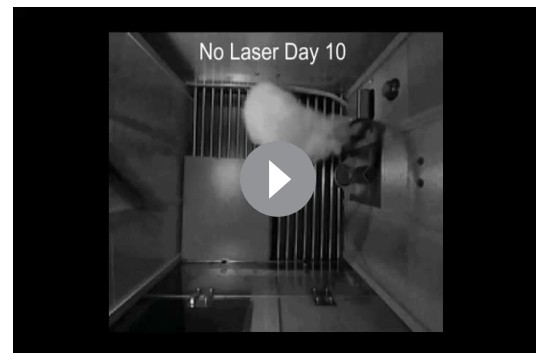

**Video 1.** 4 Hz photoactivation of rostral PL neurons during the tone impairs avoidance. Video of an individual rat with ChR2 infused into rPL showing avoidance behavior on the last day of avoidance training (Day 10) at Tone 1, followed by the rat's behavior at Test (Day 11) with the laser on during the tone (4 Hz, 30 s duration, 5 ms pulse width, 8–10 mW light intensity).
DOI: https://doi.org/10.7554/eLife.34657.016

induced property of the tone, indicating that shock is avoidable and that the rat has the option to avoid. 'Avoidability' in this task resembles 'controllability' when rats learn that they can terminate a shock by running in a wheel (*Maier and Seligman, 1976*; *Maier, 2015*). In that task, rats' control of shock reduced the activation of serotonergic neurons in the dorsal raphe, a phenomenon blocked by PL inactivation (*Amat et al., 2005*). Thus, inhibition in PL may reduce its effects on target structures such as the raphe, thereby signaling avoidability/controllability in a variety of contexts.

We impaired avoidance by photostimulating at 4 Hz, which clamped PL glutamatergic neurons to their basal firing rate and prevented tone-induced inhibition. This rate of stimulation is much lower than the 20 Hz used in most behavioral studies employing channelrhodopsin (*Liu et al., 2012*; *Felix-Ortiz and Tye, 2014*; *Marcinkiewcz et al., 2016*; *Villaruel et al., 2017*; *Burgos-Robles et al., 2017*; *Warlow et al., 2017*). This impairment in avoidance with 4 Hz stimulation was surprising given that photoactivation of the adjacent infralimbic cortex required stimulation rates $\geq$ 10 Hz to reduce conditioned freezing (*Do-Monte et al., 2015a*). As 4 Hz approximates the average firing rate of mPFC putative projection neurons (*Jung et al., 1998*; *Baeg et al., 2001*; *Burgos-Robles et al., 2009*; *Sotres-Bayon et al., 2012*), the impairment in avoidance was likely due to abolishment of inhibitory responses. Excitatory responses coupled with the loss of inhibitory responses to the tone would cause PL responses in avoidance-trained rats to resemble those in fear conditioned rats, indicating that the shock is not avoidable. An important caveat, however, is that CaMKIIα-expressing neurons were activated by ChR2 indiscriminately and were not limited to neurons showing inhibitory responses to the tone. Thus, in addition to reducing inhibitory responses in one population of cells, we likely induced some degree of excitation in a separate population of cells. Both mechanisms would have the effect of increasing tone-induced activity at rPL targets, but ChR2 photoactivation would be expected to have a greater effect (as we observed). Whereas MUS inactivation would non-specifically inhibit all neuronal types, it may resemble our 4 Hz photoactivation by preventing any further inhibition at tone onset.

Neurons in PL project to the basolateral amygdala (BLA) and ventral striatum (VS; *Sesack et al., 1989*; *Vertes, 2004*), both necessary for active avoidance (*Darvas et al., 2011*; *Bravo-Rivera et al., 2014*; *Ramirez et al., 2015*; *Hormigo et al., 2016*). Inhibition of excitatory inputs from rPL to VS may be permissive for avoidance behavior, which would resemble inhibition of VS during food seeking (*Rada et al., 1997*; *Saulskaya and Mikhailova, 2002*; *Do-Monte et al., 2017*). rPL activity may also modulate avoidance via projections to BLA, thereby activating BLA projections to VS, which have been shown to drive shuttle avoidance (*Ramirez et al., 2015*). One possibility is that inputs to VS from PL and BLA drive different aspects of avoidance: rPL for avoidance early in the tone when it is less urgent, and BLA for avoidance later in the tone when it is more urgent. In support of this, PL inhibition often delayed but did not block avoidance, revealing the effect of other inputs to VS later in the tone.

Excessive avoidance is clinically relevant for PTSD and other anxiety disorders. Rodent PL is considered to be homologous to the human dorsal anterior cingulate cortex (dACC; *Bicks et al., 2015*; *Heilbronner et al., 2016*). In humans, active avoidance is correlated with functional coupling of the rostral dACC with the striatum (*Collins et al., 2014*), and the ability to control aversive stimuli is associated with decreased activity in the rostral dACC (*Wood et al., 2015*), consistent with the rPL inhibition we observed. Furthermore, excessive avoidance in PTSD patients is correlated with increased activity in rostral dACC (*Marin et al., 2016*). Thus, reduced inhibition in rostral dACC and its striatal targets may bias individuals toward avoidance, despite behavioral costs and a low probability of danger.

## Materials and methods

### Subjects

A total of 155 adult male Sprague Dawley rats (Harlan Laboratories, Indianapolis, IN) aged 3–5 months and weighing 320–420 g were housed and handled as previously described (*Bravo-Rivera et al., 2014*). Rats were maintained on a restricted diet (18 g/day) of standard laboratory rat chow to facilitate pressing a bar for food on a variable interval schedule of reinforcement (VI-30). All procedures were approved by the Institutional Animal Care and Use Committee of the University of

Puerto Rico School of Medicine in compliance with the National Institutes of Health guidelines for the care and use of laboratory animals.

## Surgery

Rats were anesthetized with isofluorane inhalant gas (5%) first in an induction chamber, then positioned in a stereotaxic frame (Kopf Instruments, Tujunga, CA). Isofluorane (2–3%) was delivered through a facemask for anesthesia maintenance. For pharmacological inactivations, rats were implanted with 26-gauge double guide cannulas (Plastics One, Roanoke, VA) in the prelimbic prefrontal cortex (PL; +3.0 mm AP; ±0.6 mm ML; −2.5 mm DV to bregma, 0° angle). For optogenetic experiments, rats were bilaterally implanted with 22-gauge single guide cannulas (Plastics One, Roanoke, VA) in the prelimbic prefrontal cortex (PL; +2.6–2.8 mm AP; ±1.50 mm ML; −3.40 mm DV to bregma, 15°angle). An injector extending 2 mm beyond the tip of each cannula was used to infuse 0.5 µl of virus at a rate of 0.05 µl/min. The injector was kept inside the cannula for an additional 10 min to reduce back-flow. The injector was then removed and an optical fiber (0.22 NA, 200 nm core, constructed with products from Thorlabs, Newton, NJ) with 1 mm of projection beyond the tip of each cannula was inserted for PL illumination. The guide cannula and the optical fiber were cemented to the skull (C and B metabond, Parkell, Brentwood, NY; Ortho Acrylic, Bayamón, PR). For unit recording experiments, rats were implanted with a moveable array of 9 or 16 microwires (50 µm spacing, 3 × 3 or 2 × 8, Neuro Biological Laboratories, Denison, TX) targeting regions of PL along the rostral-caudal axis. After surgery, triple antibiotic was applied topically around the surgery incision, and an analgesic (Meloxicam, 1 mg/Kg) was injected subcutaneously. Rats were allowed a minimum of 7 days to recover from surgery prior to behavioral training.

## Behavior

Rats were initially trained to press a bar to receive food pellets on a variable interval reinforcement schedule (VI-30) inside standard operant chambers (Coulbourn Instruments, Whitehall, PA) located in sound-attenuating cubicles (MED Associates, St. Albans, VT). Bar-pressing was used to maintain a constant level of activity against which avoidance and freezing could reliably be measured. Rats were trained until they reached a criterion of ≥15 presses/min. Rats pressed for food throughout all phases of the experiment.

For platform-mediated avoidance, rats were trained as previously described (*Bravo-Rivera et al., 2014*). Briefly, rats were conditioned with a pure tone (30 s, 4 kHz, 75 dB) co-terminating with a scrambled shock delivered through the floor grids (2 s, 0.4 mA). The inter-trial interval was variable, averaging 3 min. An acrylic square platform (14.0 cm each side, 0.33 cm tall) located in the opposite corner of the sucrose pellet–delivering bar protected rats from the shock. The platform was fixed to the floor and was present during all stages of training (including bar-press training). Rats were conditioned for 10 days, with nine tone-shock pairings per day with a VI-30 schedule maintained across all training and test sessions. The availability of food on the side opposite to the platform motivated rats to leave the platform during the inter-trial interval, facilitating trial-by-trial assessment of avoidance. Once rats learned platform-mediated avoidance, rats underwent a 2-tone expression test (two tones with no shock). Tone 2 served as an unstimulated within-subject control and was included in the experimental design to identify any persistent effects of the laser activation. In all optogenetic experiments, the response to Tone 1 was statistically compared to the eYFP control group at Tone 1.

## Drug infusions

The GABA-A agonist muscimol (fluorescent muscimol, BODIPY TMR-X conjugate, Sigma-Aldrich) was used to enhance GABA-A receptor activity, thereby inactivating target structures. Infusions were made 45 min before testing at a rate of 0.2 µl/min (0.11 nmol/ 0.2 µl/ per side), similar to our previous studies (*Do-Monte et al., 2015b*; *Rodriguez-Romaguera et al., 2016*).

## Viruses

The adeno-associated viruses (AAVs; serotype 5) were obtained from the University of North Carolina Vector Core (Chapel Hill, NC). Viral titers were $4 \times 10^{12}$ particles/ml for channelrhodopsin (AAV5:CaMKIIα::hChR2(H134R)-eYFP) and archaerhodopsin (AAV5:CaMKIIα::eArchT3.0-eYFP) and 3

× 10$^{12}$ particles/ml for control (AAV5:CaMKIIα::eYFP). Rats expressing eYFP in PL were used to control for any nonspecific effects of viral infection or laser heating. The CaMKIIα promoter was used to enable transgene expression favoring pyramidal neurons (*Liu and Jones, 1996*) in cortical regions (*Jones et al., 1994*; *Van den Oever et al., 2013*; *Warthen et al., 2016*). Viruses were housed in a −80°C freezer until the day of infusion.

## Laser delivery

Rats expressing channelrhodopsin (ChR2) in PL were illuminated using a blue diode-pump solid state laser (DPSS, 473 nm, 2 or 4 Hz, 5 ms pulse width, 8–10 mW at the optical fiber tip; OptoEngine, Midvale, UT), similar to our previous study (*Do-Monte et al., 2015a*). Rats expressing archaerhodopsin (ArchT) in PL were bilaterally illuminated using a DPSS green laser (532 nm, constant, 10–12 mW at the optical fiber tip; OptoEngine). For both ChR2 and ArchT experiments, the laser was activated at tone onset and persisted throughout the 30 s tone presentation. Laser light was passed through a shutter/coupler (200 nm, Oz Optics, Ontario, Canada), patchcord (200 nm core, ThorLabs, Newton, NJ), rotary joint (200 nm core, 2 × 2, Doric Lenses, Quebec city, Canada), dual patchcord (0.22 NA, 200 nm core, ThorLabs), and bilateral optical fibers (made in-house with materials from ThorLabs and Precision Fiber Products, Milpitas, CA) targeting the specific subregions in PL. Rats were familiarized with the patchcord during bar press training and during the last 4 d of avoidance training before the expression test.

## Single-unit recordings

Rats implanted with moveable electrode arrays targeting PL/Cg1 were either avoidance conditioned as previously described or exposed to the training environment (platform, tone presentations, behavior box) in the absence of the shock. Extracellular waveforms that exceeded a voltage threshold were digitized at 40 kHz and stored on a computer. Waveforms were then sorted offline using three-dimensional plots of principal component and voltage vectors (Offline Sorter; Plexon, Dallas, TX) and clusters formed by individual neurons were tracked. Timestamps of neural spiking and flags for the occurrence of tones and shocks were imported to NeuroExplorer for analysis (NEX Technologies, Madison, AL). Because we used a high impedance electrode in the current study (~750–1000 kOhm), we were unable to sample interneurons. Single units were recorded across the extent of Cg1, Cg2, and PL. We excluded any units in Cg2 based on histological verification. Portions of Cg1 dorsal to rPL were grouped together for analyses, and portions of Cg1 dorsal to cPL were grouped together for analyses, ensuring that the proportion of Cg1 units was similar across both PL regions. Data was recorded during the entire session except during the 2 s shock. After conditioning, rats were tested for avoidance expression.

For avoidance assessment, rats received full conditioning sessions (with shocks) across days. Inclusion of the shock prevented extinction of avoidance. After each day, electrodes were lowered 150 µM to isolate new neurons for the following session the next day. To detect tone-elicited changes in PL activity, we assessed whether neurons changed their firing rate significantly during the first 500–1000 ms after tone onset across the first five trials. A Z-score for each 500 ms bin was calculated relative to 20 pre-tone bins of equal duration (10 s pretone). PL neurons were classified as showing excitatory tone responses if the initial bins exceeded 2.58 z's (p<0.01, two-tailed). PL neurons were classified as showing inhibitory tone responses across time if any of the initial two tone bins exceeded −1.96 Z's (p<0.05, two-tailed). A longer response latency for inhibition was chosen to take into account multi-synaptic pathways that are present in inhibitory circuits.

To detect changes in PL activity during platform entry, we employed the same procedure used for assessing tone responses. We assessed whether neurons changed their firing rate significantly during the first 500–1000 ms after platform entry. A Z-score for each 500 ms bin was calculated relative to the same pretone baseline. Heat maps of single unit data were generated with Z-scores from baseline through the 28 s after tone onset or platform entry.

To assess the relations between inhibition and avoidance on a trial-by-trial basis, we compared the latency of inhibition with the latency of platform entry. The latency of the inhibitory response to the tone was identified as the start of the first interspike interval (ISI) that was significantly longer than the average ISI in 30 s of pre-tone activity (Z > 1.65; p<0.05) recorded in all cells for each trial. We then computed the average latency of inhibition and platform entry for each cell recorded across

all the trials in which successful avoidance was observed (nine trials per session). Avoidance latency was also averaged on those trials for each cell.

## Optrode recordings

Rats expressing ArchT or ChR2 in PL were anesthetized with urethane (1 g/Kg, i.p.; Sigma Aldrich) and mounted in a stereotaxic frame. An optrode consisting of an optical fiber surrounded by 8 or 16 single-unit recording wires (Neuro Biological Laboratories) was inserted and aimed at PL (AP, +2.8 mm; ML: −0.5; DV: −3.5). The optrode was ventrally advanced in steps of 0.03 mm. Single-units were monitored in real time (RASPUTIN, Plexon). After isolating a single-unit, a 532 nm laser was activated for 10 s within a 20 s period, at least 10 times for ArchT-infected PL neurons. For ChR2-infected PL neurons, a 473 nm laser was activated for 30 s at a rate of 2 or 4 Hz (5 ms pulse width) within a 90 s period (60 s ITI), at least five times. Single-units were recorded and stored for spike sorting (Offline Sorter, Plexon) and spike-train analysis (Neuorexplorer, NEX Technologies). Excitatory and inhibitory responses were calculated by comparing the average firing rate of each neuron during the 10 s of laser OFF with the 10 s of laser ON for ArchT neurons and during 30 s laser OFF just prior to the 30 s of laser ON for ChR2 neurons (Wilcoxon signed-rank test, 1 s bins).

## Open field task

Locomotor activity in the open field arena (90 cm diameter) was automatically assessed (ANY-Maze) by comparing the total distance travelled between 30 s trials (laser off versus laser on), following a 3 min acclimation period for optogenetic experiments. The distance traveled was used to assess locomotion and time in center was used to assess anxiety. For pharmacological inactivation experiments, distance traveled and time in center was measured over a 5 min period following a 3 min acclimation period 45 min after MUS or SAL was infused prior to sacrificing animals.

## Histology

After behavioral experiments, rats were deeply anesthetized with sodium pentobarbital (450 mg/kg i.p.) and transcardially perfused with 0.9% saline followed by a 10% formalin solution. Brains were removed from the skull and stored in 30% sucrose for cryoprotection for at least 72 hr before sectioning and Nissl staining. Histology was analyzed for placement of cannulas, virus expression, and electrodes.

## Data collection and analysis

Behavior was recorded with digital video cameras (Micro Video Products, Peterborough, Ontario, Canada). Freezing and platform avoidance was quantified by observers blind to the experimental group. Freezing was defined as the absence of all movement except for respiration. Avoidance was defined as the rat having at least three paws on the platform. We calculated percent suppression of bar pressing for each tone as previously described (*Bravo-Rivera et al., 2014*):

$$\frac{(pretone\ rate - tone\ rate)}{(pretone\ rate + tone\ rate)} * 100$$

A value of 0% indicates no suppression, where a value of 100% indicates complete suppression. To calculate pretone rates, we used the 60 s before tone onset. In a subset of animals, AnyMaze software was available for recording and calculating freezing, avoidance, and suppression of bar pressing (Stoelting, Wood Dale, IL). The time spent avoiding during the tone (percent time on platform) was used as our avoidance measure. Avoidance and freezing to the tone was expressed as a percentage of the 30 s tone presentation. Our experimental groups typically consisted of approximately 15 animals. This is typical of other laboratories and results in sufficient statistical confidence. Moreover, it also agrees with the theoretical minimum sample size given by:

$$n = \frac{z^2 \sigma^2}{d^2}$$

where z = the level of confidence desired (in standard deviations), σ = the estimate of the population standard deviation, and d = the acceptable width of the confidence interval. Technical replications, testing the same measurement multiple times, and biological replications, performing the same test

on multiple samples (individual rats or single units), were used to test the variability in each experiment. Statistical significance was determined with Student's two-tailed t-tests, Fisher Exact tests, Chi Square tests, Pearson's correlation, Mann Whitney U tests, or repeated-measures ANOVA, followed by post hoc Tukey analyses, and Bonferroni corrections, where appropriate using STATISTICA (Statsoft, Tulsa, OK) and Prism (Graphpad, La Jolla, CA).

## Acknowledgements

This study was supported by NIH grants F32-MH105185 to MMD, R36-MH102968 to CBR, R36-MH105039 to JRR, R37-MH058883 and P50-MH106435 to GJQ, and the University of Puerto Rico President's Office. We thank Drs. Denis Pare and Drew Headley for comments on an earlier version. We also thank Valeria Lozada-Miranda, Joyce Mendoza-Navarro, Jorge Iravedra-Garcia, Fabiola Gonzalez-Díaz, and Jorge Maldonado de Jesus for help with behavioral experiments, Mark Diltz, Ethan Faryna, and Ladik Fernandez for help with data analysis, Carlos Rodríguez and Zarkalys Quintero for technical assistance, Dr. Karl Deisseroth for viral constructs, and the UNC Vector Core Facility for viral packaging.

## Additional information

### Funding

| Funder | Grant reference number | Author |
| --- | --- | --- |
| National Institute of Mental Health | F32-MH105185 | Maria M Diehl |
| University of Puerto Rico President's Office | | Gregory J Quirk |
| National Institute of Mental Health | R36-MH102968 | Christian Bravo-Rivera |
| National Institute of Mental Health | R36-MH105039 | Jose Rodriguez-Romaguera |
| National Institute of Mental Health | R37-MH058883 | Gregory J Quirk |
| National Institute of Mental Health | P50-MH106435 | Gregory J Quirk |

The funders had no role in study design, data collection and interpretation, or the decision to submit the work for publication.

### Author contributions

Maria M Diehl, Conceptualization, Data curation, Software, Formal analysis, Supervision, Funding acquisition, Validation, Investigation, Visualization, Methodology, Writing—original draft, Project administration, Writing—review and editing; Christian Bravo-Rivera, Conceptualization, Data curation, Formal analysis, Funding acquisition, Investigation, Visualization, Methodology, Writing—original draft, Project administration, Writing—review and editing; Jose Rodriguez-Romaguera, Conceptualization, Data curation, Funding acquisition, Validation, Investigation, Visualization, Methodology, Writing—review and editing; Pablo A Pagan-Rivera, Validation, Investigation, Visualization, Writing—original draft, Performed all MUS experiments and contributed to data visualization and analysis for Figure 1, Contributed to data collection for ArchT experiments; Anthony Burgos-Robles, Conceptualization, Data curation, Methodology, Writing—review and editing, Contributed to data analysis and interpretation, Provided critical review, commentary, and revisions; Ciorana Roman-Ortiz, Validation, Investigation, Writing—original draft, Performed initial unit recording surgeries and experiments, Contributed to data collection and analysis for unit recordings in Figure 3; Gregory J Quirk, Conceptualization, Resources, Data curation, Formal analysis, Supervision, Funding acquisition, Validation, Visualization, Methodology, Writing—original draft, Project administration, Writing—review and editing

## Author ORCIDs

Maria M Diehl ⬥ http://orcid.org/0000-0002-7370-6106
Anthony Burgos-Robles ⬥ https://orcid.org/0000-0002-6729-2648
Gregory J Quirk ⬥ http://orcid.org/0000-0002-7534-2764

## Ethics

Animal experimentation: This study was performed in strict accordance with the recommendations in the Guide for the Care and Use of Laboratory Animals of the National Institutes of Health. All of the animals were handled according to approved institutional animal care and use committee (IACUC) protocols (#A3340107) of the University of Puerto Rico. All surgery was performed under isoflurane anesthesia, and every effort was made to minimize suffering.

## Decision letter and Author response

Decision letter https://doi.org/10.7554/eLife.34657.022
Author response https://doi.org/10.7554/eLife.34657.023

## Additional files

### Supplementary files

• Transparent reporting form
DOI: https://doi.org/10.7554/eLife.34657.017

### Data availability

All data generated or analysed during this study are included in the manuscript and supporting files. Source data files have been provided for all figures.

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
