## [Decision Letter]

Thank you for submitting your article "Active avoidance requires inhibitory signaling in the rodent prelimbic prefrontal cortex" for consideration by *eLife*. Your article has been reviewed by three peer reviewers, one of whom, Geoffrey Schoenbaum, is a member of our Board of Reviewing Editors, and the evaluation has been overseen by Michael Frank as the Senior Editor. Michael McDannald has agreed to reveal his identity as one of the other reviewers.

The reviewers have discussed the reviews with one another and the Reviewing Editor has drafted this decision to help you prepare a revised submission.

Summary:

This study examines the role of the prelimibic cortex (PL) in active avoidance. The authors use fluorescent-tagged muscimol to extend a previous finding that PL inhibition delays already established active avoidance. To the author's surprise, optogenetic PL inhibition at the time of cue in this same task had no effect on avoidance – or even sped avoidance initiation when the rostral PL was photo-inhibited. Using single-unit recording they uncovered a novel, inhibitory signal in rostral PL neurons that only emerged during tone presentation in avoidance rats. Accounting for the magnitude of firing decrease observed by these neurons, the authors show that 4 Hz photo-stimulation via channelrhodopsin at only the time of the tone – cancelling out the observed inhibitory – delayed inhibitory avoidance. Collectively the results point to an unappreciated role of inhibition in PL in controlling the initiation of the decision to avoid. The topic is timely, the results are quite novel, and technically these experiments are excellent. Further, using neural activity to calibrate the optogenetic manipulation is smart and would be a good practice for the field. Overall, this is a very strong manuscript, mostly requiring just some clarifying analyses.

Essential revisions:

In our discussion, the reviewers identified three key areas that some additional information would be helpful. I have left the comments below intact, but these three revisions are the ones that are most important to us. The first is to provide additional detail on what behavior the rats do instead of avoiding when PL is manipulated. This is interesting in its own right and also will help to identify the role of PL – for example what happens might be different depending on whether PL is actually involved in triggering the decision versus evoking fear. All three reviewers agreed this was useful, though it is described by R1. In addition, we also thought additional detail on the single unit data would be helpful. R2 would like more information regarding the excitatory correlates and their relation or lack thereof to behavior, since they are presumably affected by the inhibitory tone. R3 is interested in developing stronger correlative relationships between firing activity and the actual decision. In part, the question here is whether there is any better metric than is provided and/or if not to discuss in more detail what is meant here, given that the actual decision to avoid is so hard to define experimentally – how can you say this is really what the neurons are doing?

*Reviewer #1:*

In this study, the authors examine the roles of rostral and caudal PL in active avoidance. Rats were trained to avoid a cue-signaled shock by interrupting their ongoing responding for food and stepping onto a platform. Prior work had shown that inactivating PL disrupted this behavior without affecting the cue-induced freezing. The current study replicated this effect with muscimol and then explored the neural correlates and effects of more temporally specific optogenetic manipulations. Contrary to expectations, they found that optogenetic inhibition of PL had minimal effects on expression of the active avoidance response. Subsequent recording work suggested that this was because the activity in rPL correlated with the response was inhibitory rather than excitatory. Based on this, the authors optogenetically activated PL during the cue, modestly to counteract the normal inhibition. Activation of rPL (but not cPL) impaired avoidance selectively and completely.

This is a remarkable set of studies. The authors do an excellent job of exploring a puzzling initial result, showing as they note the importance of linking optogenetic manipulations to known firing patterns in areas of interest in relevant behaviors. A particular strength of the study is the match between the effect of the effective activation on unit activity and the size of the suppression, as well as the remarkable specificity of the findings to rostral but not caudal PL, which provides a very effective control for general effects. I was a bit confused as to what the precise role of the PL contribution was to avoidance and how this related to its role in fear/freezing, which led to the following suggestions.

1) Probably the most important one is that I'd like to see the alternative behaviors plotted in detail similar to the one of interest – platform time. While I do not doubt the specificity of the role of rPL, it remains a question from the current data whether disrupting this function makes the rats no longer fearful or fearful but just incapable of organizing the escape response. There is an increase in freezing in one experiment but no effect I think in a later experiment…… but the freezing is very low. Does this mean that the rats continue to barpress for food? This question can be resolved simply by comparing the different behaviors across time – freezing, barpressing, platform. As they can be defined to be mutually exclusive, this more complete analysis may be quite interesting.

2) A second related minor suggestion – which I understand may not be worth doing – is that it would be interesting to see what happens to the response if the rPL is activated later in the cue, when the rat is on the platform. That is, in the final experiment, the rPL was activated from the start of the cue, and the escape response was essentially put on hold until the stimulation was terminated. This suggests that the area is triggering the response to avoid. But does it have an ongoing role? This could be addressed by waiting until a platform response was executed and then activating the rPL – do the rats leave the platform? And what do they do – resume barpressing? Freeze?

3) Lastly, I think the authors might provide a bit more overview as to how these results fit with the existing data regarding how PL is involved in simple fear conditioning. They seem to suggest that the entire network has been altered by the avoidance training. And yet there are still neurons that seem to signal the fear. How does the lack of effect here fit with effects in experiments lacking avoidance. Some integration with the simpler picture would be helpful.

*Reviewer #2:*

In this manuscript, Diehl and colleagues performed a set of experiments examining the role of the prelimibic cortex (PL) in active avoidance. The authors use fluorescent-tagged muscimol to extend a previous finding that PL inhibition delays already established active avoidance. To the author's surprise, optogenetic PL inhibition at the time of cue in this same task had no effect on avoidance – or even sped avoidance initiation when the rostral PL was photo-inhibited. Using single-unit recording they uncovered a novel, inhibitory signal in rostral PL neurons that only emerged during tone presentation in avoidance rats. Accounting for the magnitude of firing decrease observed by these neurons, the authors show that 4 Hz photo-stimulation via channelrhodopsin at only the time of the tone – cancelling out the observed inhibitory – delayed inhibitory avoidance. By contrast, 4 Hz stimulation outside of cue presentation and 2 Hz photo-stimulation during the same cue period had no effect on avoidance. Collectively, the results reveal a new role for the rostral PL in active avoidance that is underpinned by inhibition of firing by presumptive output neurons.

The topic is timely, and avoidance is severely understudied compared to fear conditioning. This might be due to the notorious difficulty in establishing avoidance behavior in the lab. The procedure used here is very clever and looks to produce robust avoidance behavior. Technically these experiments are excellent. Further, using neural activity to calibrate the optogenetic manipulation is smart and would be a good practice for the field. Overall, this is a very strong manuscript. My primary concern is the single-unit recording results. Actually, I am not so much concerned as I am convinced that there is a lot more information in the firing than is presented. The manuscript would be greatly improved by digging into this a bit more. Specific comments are below.

1) The authors convincingly show that the proportion of excitatory neurons recruited by tone onset does not differ between avoidance, naïve and fear groups. However, the authors do not directly compare neural activity over tone presentation between these three groups. This is a major oversight. In Figure 3D (bottom left) it is clear that while Avoidance and Naïve rats show similar proportions on at tone onset, neuron # falls off dramatically for Avoidance but not Naïve. It would be most informative to plot Z firing (as in Figure 3H) for entirety of tone presentation for the excitatory neurons of the three groups (Avoidance, Naïve and Fear). This is critical not only for understanding what these neurons are doing in the task, but for interpreting the optogenetic stimulation results. As the authors point out, the optogenetic manipulation affects both the excitatory and inhibitory populations. For this reason, it is essential to see the full 28-s excitatory response (as is shown for the inhibitory response in Figure 4A). ANOVA with factors of time and group would be capable of revealing differential firing (main effect of group or group x time interaction) or supporting the author's claim of no differential firing by revealing only a main effect of time.

2) A recurring claim (e.g. subsection “Inhibitory responses in rostral PL neurons correlate with the initiation of avoidance”, subsection “Countering inhibitory responses in rostral PL neurons delays or prevents avoidance” and the Discussion section) is that initiation of avoidance is correlated with inhibitory tone responses of rostral PL neurons. This claim is based on of the observation that a significantly greater proportion of inhibitory neurons are observed in the Avoidance group, compared to Naïve and Fear. It is further shown that within this population, the majority of neurons show inhibition linked to cue onset (20/25) whereas a smaller population maintains inhibition for the cue duration (5/25). However, given the analyses performed, it would be more accurate to say that inhibitory tone responses are a correlate observed only in Avoidance rats. If the authors wish to claim that avoidance is correlated with activity, then they need to perform an analysis that directly addresses this. For example, showing that the magnitude of the firing decrease (or some other aspect of firing) predicts the latency of platform entry – percent time on platform – on a trial-by-trial basis. Finding a significant correlation would support the author's claim. However, failing to observe a significant correlation would provide more information about this signal. For that matter, failing to observe a correlation would not dampen my enthusiasm for this manuscript. Inhibitory PL neurons may signal that, within a given context, shock can be avoided. If this were the case, one would expect the population activity observed in Figure 4A, but would not necessarily expect PL activity to predict avoidance on a trial-by-trial basis.

3) The analysis and visualization methods used to show that neural activity to the tone and platform entry were not optimal. The population data shown in Figure 3H are a good start, but this is only a small part of the story. For example, the authors do not show the corresponding population activity for the platform entry responsive neurons. Even further, it is clear from Figure 4H that there was variability in the platform response by tone neurons, indicating that some when responsive to platform entry. It could be that the magnitude of the tone response predicts the magnitude of the platform response, but that the tone response was higher across the board. The authors note that activity to platform entry could simply result from a continuing response to tone. A simple way to address this would be to compare differential tone firing (tone onset – baseline) against differential platform entry firing (platform entry onset vs last 10 s of tone). This would provide a more thorough description of the relationship between these two signals.

*Reviewer #3:*

Diehl et al. is an interesting, well-designed, appropriately-controlled study that uses electrophysiology and optogenetics to provide compelling evidence for the importance of PL inhibitory activity during active avoidance. I think this manuscript will be appropriate for publication in *eLife* following relatively modest revisions and will make a valuable contribution to the literature.

1) It is not clear that the authors addressed their primary question: Is PL activity correlated with the initial decision to avoid? They showed PL inhibitory activity is associated with tone onset, and disrupting this activity pattern impairs avoidance, but I think they can't yet make the jump to this inhibitory activity being correlated with the decision-making process. Further support for this claim could potentially be gained by further analysis- e.g. looking in individual animals at whether presence/frequency of inhibitory unit activity is correlated with latency/likelihood to avoid. Alternatively, the scope of the question to be addressed could be changed so it's more limited.

2) Similarly, the authors state that inhibitory responses at tone onset correlated with avoidance initiation. However, this seems to be inferred via comparison with control groups instead of being temporally evident from their data- in fact, the data suggest the inhibitory responses are not correlated with avoidance initiation (e.g. Figure 3G). Please clarify. On a related note, it is difficult to confidently assign the inhibitory responses to avoidance initiation, since it's not clear what that behavior would look like- the first movement to the platform? a mental decision with no observable behavioral correlate? cessation of lever pressing? are we looking at the correlate of one of these other behaviors, instead of avoidance initiation? This can potentially be addressed via examination of behavioral correlates of single unit activity at tone onset if they have the data, or by limiting the interpretation.

3) Rats were included whose viral expression spread into Cg1, due to the rationale that Cg1 and PL perform similar functions. While this has been shown in fear conditioning (Courtin et al., 2014), it has not been examined in active avoidance. More discussion of this is warranted, since many of the caudal PL recording placements in Figure 3A are in Cg1, not cPL.

4) The authors optogenetically stimulate or inhibit PL neurons during the first of two extinction tones, but the rationale is not discussed. If Tone 2 is meant to serve as an unstimulated within-subject control, they should discuss potential confounding factors that could affect behavior during Tone 2 (short-term plasticity, rebound excitation, etc.)

5) Rationale for stimulating at 4Hz instead of 6Hz isn't clear. Is there evidence to indicate optogenetic stimulation at a certain frequency has an additive effect on a cell's current frequency, rather than causing entrainment at the stimulation frequency?

6) The authors should discuss differences between rPL and cPL further and provide a stronger rationale for analyzing them independently. Where does cPL project? What might explain why rPL, not cPL, is involved in active avoidance? Is there a different breakdown in rPL and cPL in their previous cFos studies?

7) One of the major reasons for selecting the brief post-tone latency for analysis was to ensure PL neuron activity is limited to tone, and not subsequent behavior. However, this is not convincing, given there seems to be significant avoidance even at that point. Expanding the Figure 3C x-axis around the time of tone onset will help evaluate this issue.

---

## [Author Response]

Essential revisions:

In our discussion, the reviewers identified three key areas that some additional information would be helpful. I have left the comments below intact, but these three revisions are the ones that are most important to us. The first is to provide additional detail on what behavior the rats do instead of avoiding when PL is manipulated. This is interesting in its own right and also will help to identify the role of PL – for example what happens might be different depending on whether PL is actually involved in triggering the decision versus evoking fear. All three reviewers agreed this was useful, though it is described by R1. In addition, we also thought additional detail on the single unit data would be helpful. R2 would like more information regarding the excitatory correlates and their relation or lack thereof to behavior, since they are presumably affected by the inhibitory tone. R3 is interested in developing stronger correlative relationships between firing activity and the actual decision. In part, the question here is whether there is any better metric than is provided and/or if not to discuss in more detail what is meant here, given that the actual decision to avoid is so hard to define experimentally – how can you say this is really what the neurons are doing?

Reviewer #1:

[…] This is a remarkable set of studies. The authors do an excellent job of exploring a puzzling initial result, showing as they note the importance of linking optogenetic manipulations to known firing patterns in areas of interest in relevant behaviors. A particular strength of the study is the match between the effect of the effective activation on unit activity and the size of the suppression, as well as the remarkable specificity of the findings to rostral but not caudal PL, which provides a very effective control for general effects. I was a bit confused as to what the precise role of the PL contribution was to avoidance and how this related to its role in fear/freezing, which led to the following suggestions.1) Probably the most important one is that I'd like to see the alternative behaviors plotted in detail similar to the one of interest – platform time. While I do not doubt the specificity of the role of rPL, it remains a question from the current data whether disrupting this function makes the rats no longer fearful or fearful but just incapable of organizing the escape response. There is an increase in freezing in one experiment but no effect I think in a later experiment…… but the freezing is very low. Does this mean that the rats continue to barpress for food? This question can be resolved simply by comparing the different behaviors across time – freezing, barpressing, platform. As they can be defined to be mutually exclusive, this more complete analysis may be quite interesting.

The reviewer logically asks, “what is the rat doing if it is not avoiding?” when rPL is manipulated. To address this, we further assessed freezing and suppression of bar pressing when PL was manipulated with MUS, ARCH, or ChR2. Neither freezing nor suppression were significantly reduced by any of the manipulations, indicating that the reduction in avoidance was not accompanied by a reduction in fear. We have added insets to all the behavioral figures (Figure 1, Figure 2 and Figure 6) showing freezing and suppression of bar pressing. As we had originally reported, MUS in PL actually increased freezing during the tone (see inset to Figure1). Analysis of 3-second bins revealed that this effect reached significance only at seconds 18-21 (repeated measures ANOVA, post hoc Tukey, panel A below), making it unlikely that the reduction in avoidance early in the tone was due to increased freezing. A similar 3-second bin analysis of freezing for ARCH (B- rostral PL, C- caudal PL) or ChR2 (D- rostral PL, E- caudal PL) is in Author response image 1.

2) A second related minor suggestion – which I understand may not be worth doing – is that it would be interesting to see what happens to the response if the rPL is activated later in the cue, when the rat is on the platform. That is, in the final experiment, the rPL was activated from the start of the cue, and the escape response was essentially put on hold until the stimulation was terminated. This suggests that the area is triggering the response to avoid. But does it have an ongoing role? This could be addressed by waiting until a platform response was executed and then activating the rPL – do the rats leave the platform? And what do they do – resume barpressing? Freeze?

This is an interesting question that would tell us if PL signaling is needed for the maintenance of avoidance behavior. Our single-unit data demonstrate that most of the inhibitory responses at platform entry terminated soon after entry (see Figure 3—figure supplement 1C), suggesting that sustained inhibition may not be necessary for maintaining avoidance.

3) Lastly, I think the authors might provide a bit more overview as to how these results fit with the existing data regarding how PL is involved in simple fear conditioning. They seem to suggest that the entire network has been altered by the avoidance training. And yet there are still neurons that seem to signal the fear. How does the lack of effect here fit with effects in experiments lacking avoidance. Some integration with the simpler picture would be helpful.

The reviewer wants to know how the current findings, that manipulations of PL have little effect on freezing, can be reconciled with previous studies of ours and others demonstrating PL’s critical role in expression of Pavlovian conditioned freezing. It is well-established that early in avoidance training, rats use Pavlovian conditioning to learn that the tone signals shock, and then switch to instrumental avoidance learning as training continues. We suggest, therefore, that PL was serving the well-established role of supporting freezing in the early stages of training. In fact, we observed that silencing rPL during the second day of avoidance conditioning significantly reduced freezing (see subsection “Photosilencing of PL glutamatergic neurons does not delay avoidance”). However, as rats learn the avoidance response, freezing decreases and is no longer susceptible to PL manipulations. We initially reported this in our 2014 MUS study (Bravo-Rivera, et al., 2014) and observed it again here with MUS and ARCH inhibition of PL. Thus, the circuit for conditioned freezing changes dramatically with avoidance training. We now emphasize this in the Discussion section.

Reviewer #2:

[…] The topic is timely, and avoidance is severely understudied compared to fear conditioning. This might be due to the notorious difficulty in establishing avoidance behavior in the lab. The procedure used here is very clever and looks to produce robust avoidance behavior. Technically these experiments are excellent. Further, using neural activity to calibrate the optogenetic manipulation is smart and would be a good practice for the field. Overall, this is a very strong manuscript. My primary concern is the single-unit recording results. Actually, I am not so much concerned as I am convinced that there is a lot more information in the firing than is presented. The manuscript would be greatly improved by digging into this a bit more. Specific comments are below.1) The authors convincingly show that the proportion of excitatory neurons recruited by tone onset does not differ between avoidance, naïve and fear groups. However, the authors do not directly compare neural activity over tone presentation between these three groups. This is a major oversight. In Figure 3D (bottom left) it is clear that while Avoidance and Naïve rats show similar proportions on at tone onset, neuron # falls off dramatically for Avoidance but not Naïve. It would be most informative to plot Z firing (as in Figure 3H) for entirety of tone presentation for the excitatory neurons of the three groups (Avoidance, Naïve and Fear). This is critical not only for understanding what these neurons are doing in the task, but for interpreting the optogenetic stimulation results. As the authors point out, the optogenetic manipulation affects both the excitatory and inhibitory populations. For this reason, it is essential to see the full 28-s excitatory response (as is shown for the inhibitory response in Figure 4A). ANOVA with factors of time and group would be capable of revealing differential firing (main effect of group or group x time interaction) or supporting the author's claim of no differential firing by revealing only a main effect of time.

The reviewer wants to know if the magnitude of excitatory tone responses across time differ between the avoidance, naïve, and fear groups. To address this issue, we examined the z-scores of the excitatory tone responses in each group across the entire tone (see Author response image 2). Using a repeated measures ANOVA, there was no main effect of Group (F_(2,55)_=1.17, p=0.316), but a main effect of time (p<0.001), and an interaction of Group x Time (F_(2,110)_=1.245, p=0.0443), which revealed that the fear conditioned group was significantly higher than the avoidance group at the first 500 ms time point (post hoc Tukey, p<0.001; see Author response image 2). Because the differences in Figure 3D (bottom left) do not reflect changes in magnitude of firing, we have added Figure 3—figure supplement 1A-B: a heat map of the average Z-scores across the entire tone of both excitatory and inhibitory tone responses as well as a graph of the average Z-score activity of all responses to tone onset recorded from the avoidance-trained rats. The heat maps show that some cells have sustained elevated firing (see top rows of excitatory tone responsive cells in Figure 3—figure supplement 1A). The group differences in proportions of excitatory responses later in the tone (Figure 3D, bottom left) may be due to signaling of other avoidance-related behaviors that are not present in naïve rats (anticipation of shock, waiting to exit the platform to continue seeking food, etc.).

**Author response image 2. respfig2:** 

2) A recurring claim (e.g. subsection “Inhibitory responses in rostral PL neurons correlate with the initiation of avoidance”, subsection “Countering inhibitory responses in rostral PL neurons delays or prevents avoidance” and the Discussion section) is that initiation of avoidance is correlated with inhibitory tone responses of rostral PL neurons. This claim is based on of the observation that a significantly greater proportion of inhibitory neurons are observed in the Avoidance group, compared to Naïve and Fear. It is further shown that within this population, the majority of neurons show inhibition linked to cue onset (20/25) whereas a smaller population maintains inhibition for the cue duration (5/25). However, given the analyses performed, it would be more accurate to say that inhibitory tone responses are a correlate observed only in Avoidance rats. If the authors wish to claim that avoidance is correlated with activity, then they need to perform an analysis that directly addresses this. For example, showing that the magnitude of the firing decrease (or some other aspect of firing) predicts the latency of platform entry – percent time on platform – on a trial-by-trial basis. Finding a significant correlation would support the author's claim. However, failing to observe a significant correlation would provide more information about this signal. For that matter, failing to observe a correlation would not dampen my enthusiasm for this manuscript. Inhibitory PL neurons may signal that, within a given context, shock can be avoided. If this were the case, one would expect the population activity observed in Figure 4A, but would not necessarily expect PL activity to predict avoidance on a trial-by-trial basis.

The reviewer raises the valid question of whether or not the inhibitory response correlates with avoidance behavior on a trial-by-trial basis. To address this, we performed a new analysis comparing the latency of inhibition with the latency of platform entry, on a trial-by-trial basis. For a given cell on a given trial, the latency of the inhibitory response to the tone was identified as the start of the first interspike interval (ISI) that was significantly longer than the average ISI in 30 seconds of pre-tone activity (Z>1.65; p<0.05). 133/205 neurons showed at least one ISI that satisfied this criterion. For 133 neurons, the correlation between the latency of inhibition and latency of platform entry was only r=0.022 (not correlated). We then computed for each cell the average latency of inhibition and platform entry. For each cell, we averaged its inhibitory latency across all the trials in which successful avoidance was observed (n=284 trials; 9 trials in each session). We also averaged the avoidance latency on those trials. The correlation between these two averages across cells was r=0.078 (also no correlation). This result is plotted in new Figure 3—figure supplement 1E-F. Figure 3—figure supplement 1E shows that the inhibitory response of the majority of cells preceded platform entry (88/133 cells) but was not correlated with platform entry. The frequency distribution shown in Figure 3—figure supplement 1F confirm that most inhibition preceded platform entry. For trials in which the rat did not avoid (n=107 trials), there was a similar latency of inhibition (see dashed line in Figure 3—figure supplement 1F). Thus, these new analyses do not support our original hypothesis that PL inhibition initiates avoidance and is more consistent with the reviewer’s suggested hypothesis: that PL inhibitory responses signal that, within a given context, footshocks can be avoided (regardless of whether the rat chooses to avoid on that trial). We thank the reviewer for suggesting this analysis and the alternative interpretation of our results. We have modified the text throughout.

3) The analysis and visualization methods used to show that neural activity to the tone and platform entry were not optimal. The population data shown in Figure 3H are a good start, but this is only a small part of the story. For example, the authors do not show the corresponding population activity for the platform entry responsive neurons. Even further, it is clear from Figure 4H that there was variability in the platform response by tone neurons, indicating that some when responsive to platform entry. It could be that the magnitude of the tone response predicts the magnitude of the platform response, but that the tone response was higher across the board. The authors note that activity to platform entry could simply result from a continuing response to tone. A simple way to address this would be to compare differential tone firing (tone onset – baseline) against differential platform entry firing (platform entry onset vs last 10 s of tone). This would provide a more thorough description of the relationship between these two signals.

To more clearly indicate whether neurons were responsive to tone onset, platform entry or both, we removed the original population graph in 3H, and now show a Venn diagram indicating the number of neurons that overlapped across response categories (new Figure 3H). Only a small proportion of excitatory cells were responsive to both tone onset and platform entry, suggesting that these were generated by separate groups of cells. However, the inhibitory responses show more overlap between tone onset and platform entry responses: out of 22 inhibitory tone responsive and 16 inhibitory entry responsive, 9 cells were responsive to both events. The suggestion by the reviewer of comparing platform entry responses to activity during the last 10 seconds of the tone may not reveal any differences because PL activity at the end of the tone may reflect other factors such as tone duration, anticipation of shock, anticipation of platform exit to resume food-seeking, and is therefore not a reliable baseline.

Reviewer #3:

Diehl et al. is an interesting, well-designed, appropriately-controlled study that uses electrophysiology and optogenetics to provide compelling evidence for the importance of PL inhibitory activity during active avoidance. I think this manuscript will be appropriate for publication in eLife following relatively modest revisions and will make a valuable contribution to the literature.1) It is not clear that the authors addressed their primary question: Is PL activity correlated with the initial decision to avoid? They showed PL inhibitory activity is associated with tone onset, and disrupting this activity pattern impairs avoidance, but I think they can't yet make the jump to this inhibitory activity being correlated with the decision-making process. Further support for this claim could potentially be gained by further analysis- e.g. looking in individual animals at whether presence/frequency of inhibitory unit activity is correlated with latency/likelihood to avoid. Alternatively, the scope of the question to be addressed could be changed so it's more limited.

The reviewer raises the valid question of whether or not the inhibitory response correlates with avoidance behavior on a trial-by-trial basis. To address this, we performed a new analysis comparing the latency of inhibition with the latency of platform entry, on a trial-by-trial basis. For a given cell on a given trial, the latency of the inhibitory response to the tone was identified as the start of the first interspike interval (ISI) that was significantly longer than the average ISI in 30 seconds of pre-tone activity (Z>1.65; p<0.05). 133/205 neurons showed at least one ISI that satisfied this criterion. For 133 neurons, the correlation between the latency of inhibition and latency of platform entry was only r=0.022 (not correlated). We then computed for each cell the average latency of inhibition and platform entry. For each cell, we averaged its inhibitory latency across all the trials in which successful avoidance was observed (n=284 trials; 9 trials in each session). We also averaged the avoidance latency on those trials. The correlation between these two averages across cells was r=0.078 (also no correlation). This result is plotted in new Figure 3—figure-supplement 1E-F. Figure 3—figure supplement 1E shows that the inhibitory response of the majority of cells preceded platform entry (88/133 cells) but was not correlated with platform entry. The frequency distribution shown in Figure 3—figure supplement 1F confirm that most inhibition preceded platform entry. For trials in which the rat did not avoid (n=107 trials), there was a similar latency of inhibition (see dashed line in Figure 3—figure supplement 1F).

2) Similarly, the authors state that inhibitory responses at tone onset correlated with avoidance initiation. However, this seems to be inferred via comparison with control groups instead of being temporally evident from their data- in fact, the data suggest the inhibitory responses are not correlated with avoidance initiation (e.g. Figure 3G). Please clarify. On a related note, it is difficult to confidently assign the inhibitory responses to avoidance initiation, since it's not clear what that behavior would look like- the first movement to the platform? a mental decision with no observable behavioral correlate? cessation of lever pressing? are we looking at the correlate of one of these other behaviors, instead of avoidance initiation? This can potentially be addressed via examination of behavioral correlates of single unit activity at tone onset if they have the data, or by limiting the interpretation.

See response to your point #1 above. Figure 3G show responses at platform entry – the time at which the rat’s head entered the platform zone after the action to avoid had already commenced. We therefore searched for behavioral correlates that might precede platform entry, focusing on the first headturn after tone onset – which is the first movement the rat makes before proceeding to the platform. We compared the average inhibition latency with average the time of headturn for each cell showing inhibition, shown in Figure 3—figure supplement 1E-F. This analysis revealed similar results: 82/133 cells showed inhibition prior to headturn, but there was no correlation between inhibition latency and headturn latency (r=0.054).

3) Rats were included whose viral expression spread into Cg1, due to the rationale that Cg1 and PL perform similar functions. While this has been shown in fear conditioning (Courtin et al., 2014), it has not been examined in active avoidance. More discussion of this is warranted, since many of the caudal PL recording placements in Figure 3A are in Cg1, not cPL.

The reviewer wants to know if Cg1 and PL have a similar function during avoidance, as our PL viral expression reaches into Cg1. Previous studies in rabbits have found that neuronal firing in both the anterior cingulate cortex and PL correlate with avoidance learning, in which rabbits must activate a running wheel to prevent a tone-signaled footshock (Orona and Gabriel, 1983; Freeman and Gabriel, 1996). Regarding our single units, we re-assessed our histology and found that one of the cPL rats had all its units located in Cg1. Removing this rat reduced the number of Cg1 units in cPL, so that the percentage of Cg1 units is now more similar in rPL and cPL (~36%). We now state this in the Materials and methods section. With this rat removed, all cells showing inhibitory responses were located in rPL, with none in cPL (see revised Figure 3I-J).

4) The authors optogenetically stimulate or inhibit PL neurons during the first of two extinction tones, but the rationale is not discussed. If Tone 2 is meant to serve as an unstimulated within-subject control, they should discuss potential confounding factors that could affect behavior during Tone 2 (short-term plasticity, rebound excitation, etc.)

As the reviewer surmised, Tone 2 served as an unstimulated within-subject control. In terms of confounding factors, rebound excitation is unlikely as Tone 2 was presented 3 minutes after Tone 1. Short-term plasticity may have affected the behavioral response to Tone 2; however, in both ARCH and ChR2 experiments, the response to Tone 1 was statistically compared to the eYFP control group at Tone 1. Tone 2 is simply added to identify any persistent effects of the laser activation (none were observed). The rationale of Tone 2 is now discussed in the Materials and methods section.

5) Rationale for stimulating at 4Hz instead of 6Hz isn't clear. Is there evidence to indicate optogenetic stimulation at a certain frequency has an additive effect on a cell's current frequency, rather than causing entrainment at the stimulation frequency?

Yes, our intention was that naturally-occurring and optogenetically-induced spikes would sum, so that 4 Hz stimulation would cause a neuron normally firing at 2 Hz to fire at 6 Hz. However, as shown in Figure 4C, this was not the case. In anesthetized rats, cells firing at an average of 1.5 Hz baseline rate only increased to 3.3 Hz with 4 Hz optogenetic stimulation. This may be due to entrainment, as the reviewer suggests, or poor frequency following at the current levels we used. Prior studies have reported that entrainment tends to occur at low levels of ChR2 stimulation (Warden, et al., 2012). It is therefore difficult to predict exactly how cortical neurons will respond to ChR2 stimulation. Suffice it to say that our 4Hz stimulation significantly increased the rate of the PL neurons expressing ChR2 (see figure 4C). We now discuss this in the Results section.

6) The authors should discuss differences between rPL and cPL further and provide a stronger rationale for analyzing them independently. Where does cPL project? What might explain why rPL, not cPL, is involved in active avoidance? Is there a different breakdown in rPL and cPL in their previous cFos studies?

Very few studies have examined anatomical (Floyd et al., 2000; 2001) or functional (Parent et al., 2015) differences between rostral and caudal portions of PL. We investigated anatomical differences in projections of rPL and cPL within broader anatomical studies on prelimbic prefrontal cortex, which show that rPL projects less densely to ventral striatum compared to cPL (Sesack, 1989). With respect to midbrain structures, rPL projects more densely to vlPAG whereas cPL projects more densely to dlPAG (Floyd, et al., 2000), which could be a factor in selecting freezing vs. flight. Although our previous Fos studies have not differentiated rPL from cPL, we are currently examining our cFos findings in light of the new differences we have observed in rPL and cPL.

7) One of the major reasons for selecting the brief post-tone latency for analysis was to ensure PL neuron activity is limited to tone, and not subsequent behavior. However, this is not convincing, given there seems to be significant avoidance even at that point. Expanding the Figure 3C x-axis around the time of tone onset will help evaluate this issue.

The reviewer is concerned that some of the tone onset responses may be due to avoidance behavior since some avoidance latencies were <1 second. However, in only 25/284 (9%) trials did avoidance occur prior to 1000ms. Below is an expanded view of the Figure 3C showing the times of platform entry (black dots, Author response image 3). Because 91% of avoidance trials occurred after 1000ms, we believe it is the best criteria we can use to identify both excitatory (500ms) and inhibitory (up to 1000ms) tone responses. Some overlap may have occurred, and it is possible that these are the cells that showed responses to both tone onset and platform entry (see new Venn diagram in Figure 3H).

**Author response image 3. respfig3:**